# ANIMESHOOTER:
# A MULTI-SHOT ANIMATION DATASET FOR
# REFERENCE-GUIDED VIDEO GENERATION

## ABSTRACT

Recent advances in AI-generated content (AIGC) have significantly accelerated animation production. To produce engaging animations, it is essential to generate coherent multi-shot video clips with narrative scripts and character references. However, existing public datasets primarily focus on real-world scenarios with global descriptions, and lack reference images for consistent character guidance. To bridge this gap, we present **AnimeShooter**, a reference-guided multi-shot animation dataset. AnimeShooter features comprehensive hierarchical annotations and strong visual consistency across shots through an automated pipeline. Story-level annotations provide an overview of the narrative, including the storyline, key scenes, and main character profiles with reference images, while shot-level annotations decompose the story into consecutive shots, each annotated with scene, characters, and both narrative and descriptive visual captions. Additionally, a dedicated subset, AnimeShooter-audio, offers synchronized audio tracks for each shot, along with audio descriptions and sound sources. To demonstrate the effectiveness of AnimeShooter and establish a baseline for the reference-guided multi-shot video generation task, we introduce AnimeShooterGen, which leverages Multimodal Large Language Models (MLLMs) and video diffusion models. The reference image and previously generated shots are first processed by MLLM to produce representations aware of both reference and context, which are then used as the condition for the diffusion model to decode the subsequent shot. Experimental results show that the model trained on AnimeShooter achieves superior cross-shot visual consistency and adherence to reference visual guidance, which highlight the value of our dataset for coherent animated video generation.

## 1 INTRODUCTION

The animation industry plays a pivotal role in modern entertainment and education (Wells, 2013). Recent advances in AI-generated content (AIGC) have revolutionized animation production through automated creation of complex visual narratives. Professional animation workflows necessitate the generation of coherent multi-shot video sequences that maintain visual consistency and adhere to predefined character designs. This reveals a substantial gap stemming from three fundamental limitations in existing public video datasets (Bain et al., 2021; Yang et al., 2024a; Chen et al., 2024a; Wang et al., 2023b; Ju et al., 2024; Xiong et al., 2024; Wang et al., 2023a): (1) focus on real-world scenario with easily obtainable web video content, (2) reliance on global captions inadequate for multi-shot narration, and (3) absence of reference images essential for consistent character guidance across sequential shots.

In this paper, we present **AnimeShooter**, a reference-guided multi-shot animation dataset featuring comprehensive hierarchical annotations and strong visual consistency across consecutive shots. Story-level annotations define an overall storyline, main scene descriptions, and detailed character profiles with reference images. The entire story is then decomposed into ordered consecutive shots. For each shot, the shot-level annotation specifies scene, involved characters, and detailed visual captions in both narrative and descriptive forms. AnimeShooter-audio is a subset which offers additional annotations of synchronized audio for each shot, along with audio descriptions and sound sources. The dataset is constructed using an automated curation pipeline as shown in Figure 2: we first

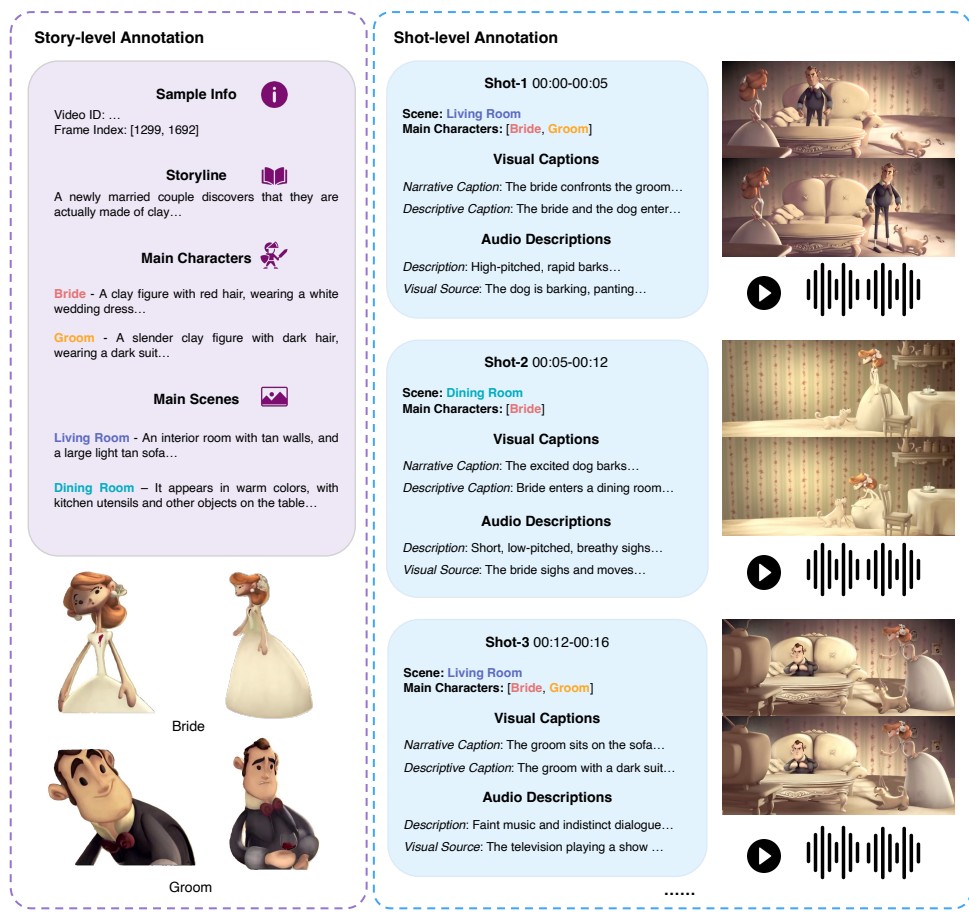

Figure 1: Overview of AnimeShooter. It is **a reference-guided multi-shot animation dataset** featuring comprehensive hierarchical annotations and strong coherence across shots. At the story level, each sample includes an overall storyline, main scene descriptions, and detailed character profiles with reference images. At the shot level, consecutive shots are annotated with specific scenes, involved characters, and rich visual captions. A specific subset, AnimeShooter-audio, additionally provides synchronized audios for each shot with corresponding audio descriptions and sound sources.

collect and filter a diverse range of large-scale animation films sourced from YouTube, then utilize Gemini (DeepMind, 2024) to generate hierarchical story scripts comprising story-level and shot-level annotations. Character reference images are extracted by sampling keyframes, segmenting characters with Sa2VA (Yuan et al., 2025) which is prompted by character ID/appearance, and ensuring quality with InternVL (Chen et al., 2024b) filtering.

To demonstrate the efficacy of AnimeShooter and establish a baseline model for this challenging task, we propose AnimeShooterGen, a reference-guided multi-shot video generation model based on MLLM and diffusion model. It can generate consecutive shots in an autoregressive manner. At each generation step, both the reference image and preceding video shots are encoded by the MLLM to produce representations that simultaneously capture character identity features and visual context. We design a multi-stage training strategy to bridge the real-to-animation domain gap and achieve autoregressive multi-shot video generation. Experiments on a custom evaluation dataset comprising multiple Intellectual Properties (IPs) and extensive evaluations demonstrate that models trained on AnimeShooter effectively learn cross-shot visual consistency and adhere to specified references.

To the best of our knowledge, this is the first reference-guided multi-shot animation dataset. Through large-scale multi-shots with visual consistency, accurate reference images for character identity, and comprehensive story and shot-level annotations, we hope AnimeShooter will facilitate research and development in narrative animation generation.

## 2 RELATED WORK

**Video-Text Datasets.** Existing text-to-video datasets present notable limitations for multi-shot animation generation. WebVid-10M (Bain et al., 2021) relies on readily available online video titles and primarily comprises short video clips. While dense-captioning datasets such as Vript (Yang et al., 2024a), ActivityNet (Caba Heilbron et al., 2015), Panda-70M (Chen et al., 2024a), and Intern-Vid (Wang et al., 2023b) offer temporally segmented clips with localized descriptions, structurally akin to multi-shot annotation via timestamp-caption pairs. However, they exhibit critical narrative deficiencies, failing to maintain coherent plot progression and suffering from temporal fragmentation with abrupt gaps or redundant overlaps. In the animation domain, while efforts like AnimeCeleb (Kim et al., 2022), Sakuga-42M (Pan, 2024), and AniSora (Jiang et al., 2024a) aim to build animation datasets, they are typically restricted to single-shot content, focus narrowly on character heads, or are not publicly available, thereby limiting their utility for multi-shot animation generation.

**Video Customization Datasets.** Video customization techniques facilitate the synthesis of videos centered on specific concepts, such as individuals or objects. A critical requirement for storytelling and animation generation is multi-shot video customization: the ability to synthesize a sequence of shots that maintain the consistent appearance of a predefined character. ID-Animator (He et al., 2024) focuses on human face synthesis, utilizing facial regions as reference images. The VideoBooth dataset (Jiang et al., 2024b), derived from WebVid (Bain et al., 2021), augments textual prompts with image prompts generated by segmenting subjects from initial video frames via Grounded-SAM (Liu et al., 2024a; Kirillov et al., 2023). Many other related efforts in video customization (Chen et al., 2025; Huang et al., 2025; Deng et al., 2025; Liu et al., 2025) with similar data construction pipelines are predominantly address single-shot synthesis, and target non-animation applications. The most related multi-shot works, MovieDreamer (Zhao et al., 2024) and MovieBench (Wu et al., 2025a), are keyframe-based and limited by dataset diversity and scale respectively.

**Multi-Shot Storytelling and Animation Generation.** Generating multi-shot videos for storytelling and animation often follows a staged pipeline. Anim-Director (Li et al., 2024) uses image generators for reference designs, which then guide keyframe generation and subsequent I2V animation. This pipeline is shared by works like VideoStudio (Long et al., 2024) and DynamiCrafter (Xing et al., 2024). Except injecting reference images via multi-modal cross-attention (e.g., Anystory (He et al., 2025), VideoStudio (Long et al., 2024)), some works attempt to maintain character appearance by optimization strategies (e.g., TaleCrafter (Gong et al., 2023), DreamRunner (Wang et al., 2024)). For example, MovieAgent (Wu et al., 2025b) leverages an LLM (Grattafiori et al., 2024; Guo et al., 2025a) for script/layout generation with ROICtrl (Gu et al., 2024) and ED-LoRA (Gu et al., 2023) for character injection. Despite these advancements, the dominant per-shot generation paradigm inherently struggles with cross-shot consistency. Recent work on Long Context Tuning (LCT) (Guo et al., 2025b) validates that autoregressive architectures can achieve enhanced holistic visual appearance and temporal coherence by recursively conditioning each shot on preceding visual contexts. But it also addresses real-world domains and faces limitations such as prohibitive training costs and a lack of explicit architectural mechanisms for reference image conditioning.

## 3 ANIMESHOOTER DATASET

This section describes the generation of AnimeShooter's structured multi-shot story script and corresponding reference images. The construction pipeline is shown in Figure 2. Please refer to supplementary files for the construction of additional subset AnimeShooter-audio.

### 3.1 DATA COLLECTION AND FILTERING

Our dataset collection begins by sourcing large-scale, diverse animated content from YouTube using keywords (e.g., "short animation", "cartoon short film"). We first filter them to ensure content relevance and minimize visual-linguistic interference. 16 uniformly sampled frames from each video are analyzed using InternVL (Chen et al., 2024b) to exclude non-animated materials (such as tutorials or film reviews) and videos containing embedded subtitles. Prolonged animated content often exhibits temporal variations in character appearance, while intricate storyline with multiple characters create cognitive overload. To mitigate these issues, we implement a duration-based filtering protocol to preserve character consistency and reduce narrative complexity. Videos exceeding 20 minutes are

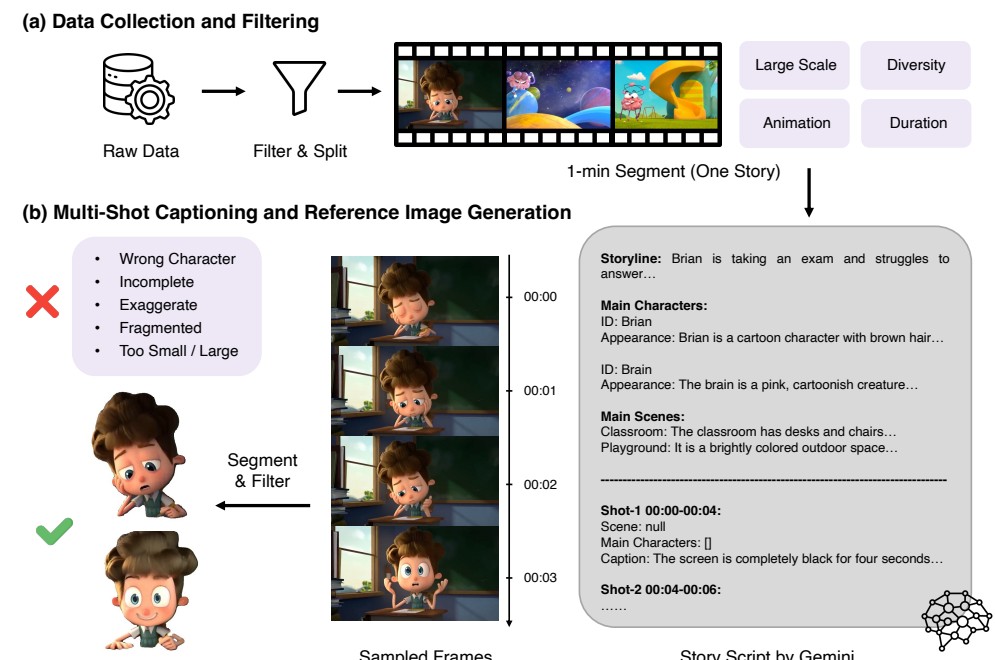

Figure 2: Video collection and annotation pipeline. We curate relevant videos from YouTube and segment them into 1-minute segments using boundary detection. Each segment serves as an individual sample representing a self-contained narrative unit (one story). We use Gemini to further decompose the story into consecutive shots with visual consistency based on transitions, and generate structured story script. Corresponding reference images are generated by Sa2VA and InternVL.

removed firstly. The remaining videos are then cut into segments with around 1-minute durations using PySceneDetect (Pys) algorithm to ensure coherent segment boundaries and narrative continuity. Each segment serves as an individual sample representing a self-contained narrative unit.

## 3.2 MULTI-SHOT CAPTIONING

To maintain the narrative cohesion and avoid referential ambiguity, we design a top-down multi-shot captioning strategy with three systematic phases: (1) Story-level annotation. This phase establishes a global narrative context by summarizing a succinct, coherent storyline and identifying 1-3 main characters and scenes, including detailed descriptions of their appearance and environment. (2) Shot decomposition. The entire story is subsequently decomposed into consecutive, non-overlapping shots delineated by shot transitions. (3) Shot-level annotation. For each shot, annotations identify the scene and characters, alongside two caption types: a narrative caption articulating plot progression (e.g., "The girl said goodbye to the bear") and a descriptive caption conveying visual details (e.g., "A girl in red standing in front of a brown bear"). We utilize Gemini-2.0-flash (DeepMind, 2024) to process 1-minute segments and generate the hierarchical story script through these three phases above.

## 3.3 REFERENCE IMAGE GENERATION

Directly extracting frames containing a specific character is an intuitive but often inadequate strategy for obtaining reference images from animated films. The frequent co-occurrence of multiple characters and the presence of complex backgrounds can significantly hinder accurate character identification and isolation. We implements a robust model-assisted segmentation and filtering workflow. The process commences by leveraging pre-extracted story scripts to retrieve all related shots. From these shots, frames are sampled at 1 fps. Candidate frames are then fed into Sa2VA (Yuan et al., 2025), which generates initial segmentation masks based on character IDs and appearance descriptions provided as text prompts. These raw masks are refined by morphological operations to fill holes and smooth contours, contour analysis to discard masks exhibiting excessive disconnected regions, and size filtering to exclude masks that occupy less than 5% or more than 90% area of the image.

Table 1: Statistics of AnimeShooter. "Num." for number, "Dur." for duration, "Chars." for characters.

| Statistics Level | Total Num. | Avg. Dur.(s) | Avg. Caption(w) | Avg. Chars. | Avg. Scenes |
|---|---|---|---|---|---|
| Video-level | 29K | 286.57 | - | - | - |
| Story-level | 148K | 56.72 | 33.55 | 2.26 | 2.20 |
| Shot-level | 2.2M | 3.85 | 41.42 | - | - |

To guarantee the final quality of reference images, InternVL (Chen et al., 2024b) performs a secondary verification, enforcing structural completeness of the segmented character, semantic coherence between the segmented region and the provided ID/appearance prompts, consistent appearance across instances relative to the character's textual description, avoidance of frames with extreme poses or expressions, and maintenance of high image resolution without motion blur.

## 3.4 DATASET STATISTICS

To ensure annotation fidelity within automated pipeline, we integrate human verification checkpoints on a small subset, validiting the accuracy of story scripts and reference images. The statistical overview of the AnimeShooter dataset is presented in Table 1. The dataset contains 29K videos, each with an average duration of 286.57 seconds. Videos are typically divided into 5.07 segments. Each segment is approximately one minute long and serves as an individual sample representing one story. These story units average 56.72 seconds and feature an average of 2.26 main characters, 2.2 main scenes, and 14.82 shots. Each shot averages 3.85 seconds and is enriched with both a 10.62-word narrative caption and a 30.8-word descriptive caption, summing to 41.42 words.

## 4 METHOD

To validate the utility of AnimeShooter and establish a baseline model for animation generation, we introduce AnimeShooterGen. Inspired by prior works (Xiang et al., 2024; Huang et al., 2024a; Zhao et al., 2024), AnimeShooterGen operates in an autoregressive fashion for reference-guided multi-shot video generation. To further augment the immersive quality, we integrate AnimeShooterGen with Text-to-Audio (TTA) model TangoFlux (Hung et al., 2024). As audio generation remains outside the scope of this paper, please refer to Appendix D for details.

## 4.1 MODEL DESIGN OF ANIMESHOOTERGEN

Given the character reference image $I_{\text{ref}}$, the previous context of the story and a natural language caption for the current shot, AnimeShooterGen predicts the current $i$-th video shot, denoted as $S_i$. Figure 3 gives an overview of the model architecture. The model has two core components: the autoregressive backbone stemming from a pretrained MLLM (Liu et al., 2024b) and a video generator based on pretrained MMDiT (Hong et al., 2022). An adapter (Q-Former (Li et al., 2023)) is added to stitch these two components. For the generation of $S_i$, the MLLM backbone $f_{\text{MLLM}}$ first processes a set of inputs: a reference image $I_{\text{ref}}$ provided by user, the accumulated previous context $C_{<i}$, and the textual caption $T_i$ for the current shot. The previous context $C_{<i}$ encapsulates the long-term memory from preceding shots and is composed of a visual context $V_{<i}$ and a textual context $\mathcal{T}_{<i}$:

$$V_{<i} = \{F_{j,\text{end}} \mid j = 1, \ldots, i-1\} \tag{1}$$

$$\mathcal{T}_{<i} = \{T_j \mid j = 1, \ldots, i-1\} \tag{2}$$

where $F_{j,\text{end}}$ represents the last frame of the previously generated $j$-th shot $S_j$, and $T_j$ is its corresponding caption. We set a sequence of learnable queries as the input of MLLM, and generate a conditioning signal $\text{Cond}_i$:

$$\text{Cond}_i = f_{\text{MLLM}}(I_{\text{ref}}, C_{<i}, T_i) \tag{3}$$

This $\text{Cond}_i$ effectively combines character visual cues from $I_{\text{ref}}$, long-term memory from $C_{<i}$, and current textual guidance from $T_i$. Subsequently, the video generator synthesizes the current shot $S_i$, and the training objective can be formulated as follows:

$$\min_{\theta} \mathbb{E}_{t,x_0 \sim p_{\text{data}}, \epsilon \sim \mathcal{N}(0,I)} \|\epsilon - \epsilon_\theta(x_t, \text{Cond}_i)\|_2^2 \tag{4}$$

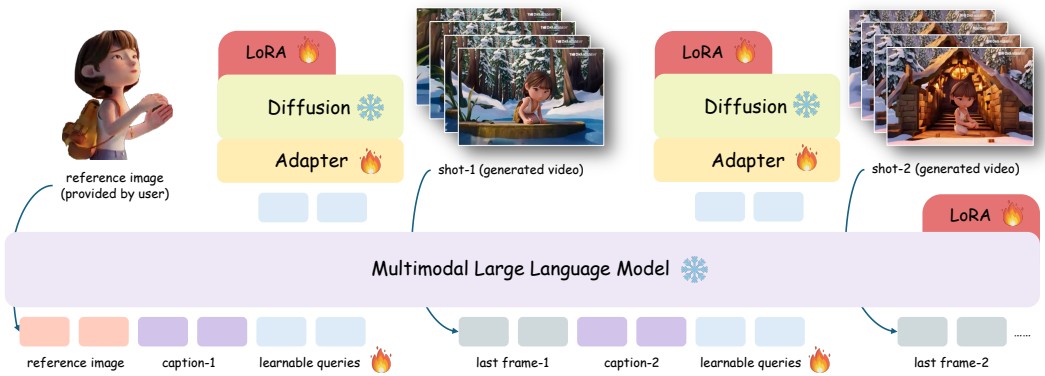

Figure 3: Overview of the model architecture. The two core components include the autoregressive backbone stemming from pretrained MLLM, and a video generator initialized from a pretrained DiT. To stitch these two components, we add a Q-Former as the adapter. This framework can generate multi-shot video in autoregressive manner.

Table 2: Quantitative comparisons of automatic metrics.

| Model | Metric | Shot-level | | | | Story-level | |
|---|---|---|---|---|---|---|---|
| | | Shot-1 | Shot-2 | Shot-3 | Shot-4 | Mean | HarMeanP |
| IP-Adapter + I2V | | 0.8004 | 0.7814 | 0.7891 | 0.7947 | 0.7914 | 0.5901 |
| Cogvideo-LoRA | CLIP ↑ | 0.7297 | 0.7200 | 0.7417 | 0.7413 | 0.7332 | 0.5028 |
| AnimeShooterGen | | **0.8022** | **0.7949** | **0.7970** | **0.7986** | **0.7982** | **0.6121** |
| IP-Adapter + I2V | | 0.3679 | 0.4169 | 0.4047 | 0.3870 | 0.3941 | 0.6818 |
| Cogvideo-LoRA | DreamSim ↓ | 0.4777 | 0.5060 | 0.4842 | 0.4864 | 0.4886 | 0.7759 |
| AnimeShooterGen | | **0.3484** | **0.3820** | **0.3799** | **0.3764** | **0.3717** | **0.6413** |

During the inference stage, upon the successful generation of shot $S_i$, its last frame $F_{i,\text{end}}$ and its caption $T_i$ are incorporated into the previous context to form $C_{<i+1} = (V_{<i} \cup \{F_{i,\text{end}}\}, \mathcal{T}_{<i} \cup \{T_i\})$, which is then used for generating the subsequent shot $S_{i+1}$. This autoregressive update mechanism allows the model to maintain coherence and narrative flow across multiple shots.

## 4.2 STAGED TRAINING

The training of AnimeShooterGen is conducted in a multi-stage fashion. The detailed training strategies and implementation details can be found in supplementary files.

**Condition Alignment:** The initial stage focuses on aligning the MLLM's output conditioning signal with the text encoder of the pretrained diffusion model. MLLM processes the first frame of a ground truth video clip and corresponding caption to generate an MLLM condition. We then minimize the MSE loss between this MLLM condition and the embedding of the caption extracted from the diffusion model's text encoder. In this stage, only the adapter and learnable queries are trainable.

**Single-Shot Training**: This stage aims to bridge the real-to-animation domain gap, and train MLLM to extract character visual attributes from the reference image. MLLM receives $I_{\text{ref}}$ and $T_i$ as input, and is optimized to produce an effective condition for generating the target shot $S_i$. In this stage, LoRA weights of MLLM, the adapter, and learnable queries are trainable.

**Multi-Shot Training**: To foster consistency across multiple shots in terms of visual appearance, style, and color palettes, this stage extends the training to sequences. MLLM now processes the reference image $I_{\text{ref}}$, the current caption $T_i$, and the previous context $C_{<i}$. LoRA weights of MLLM, the adapter, and learnable queries are trainable.

**LoRA Enhancement**: In all preceding stages, the core diffusion model remains frozen. To further enhance the character and style consistency provided by MLLM and refine overall video quality, this final stage involves test-time finetuning. Given a few video clips from a particular IP, we freeze all other model components and exclusively train LoRA weights added to the diffusion model.

Table 3: Quantitative comparisons of MLLM evaluation and user studies.

| Model | OQ ↑ | | | CRC ↑ | | | MSC ↑ | | | MCC ↑ | | |
|---|---|---|---|---|---|---|---|---|---|---|---|---|
| | GPT | Gem. | Hum. | GPT | Gem. | Hum. | GPT | Gem. | Hum. | GPT | Gem. | Hum. |
| IP-Adapter + I2V | 6.76 | 6.15 | 2.36 | 7.19 | 5.44 | 2.26 | 6.53 | 6.66 | 3.18 | 6.07 | 5.51 | 2.71 |
| Cogvideo-LoRA | 6.96 | 4.82 | 2.83 | 6.82 | 2.57 | 2.31 | 6.64 | 4.50 | 2.75 | 6.30 | 3.64 | 2.39 |
| AnimeShooterGen | **7.19** | **6.88** | **4.23** | **7.87** | **6.54** | **4.72** | **7.15** | **8.24** | **4.63** | **6.68** | **7.07** | **4.52** |

## 5 EXPERIMENTS

### 5.1 EXPERIMENT SETTING

**Baselines:** We compare two mainstream methods in storytelling field. The first approach employs a short video generation model capable of IP customization to produce individual video shots. To ensure a fair comparison and highlight the benefits of our multi-shot framework, we finetune the same pretrained diffusion model, CogVideo-2B (Hong et al., 2022), on the same IP-specific dataset. The second approach first generates a series of IP-consistent keyframes, which are then transformed into video shots using an I2V model. Keyframe generation employs SDXL (Podell et al., 2023), augmented with IP-Adapter (Ye et al., 2023) to integrate reference image features, and CogVideo-5B is utilized for the I2V conversion.

**Evaluation Dataset:** We collect 20 animation films with distinct IPs. For each IP, we manually annotate 5-6 short clips for model fine-tuning. To evaluate multi-shot generation performance, we employ DeepSeek (Guo et al., 2025a) to generate 10 unique narrative prompts per IP. Each prompt describes a story comprising 4 coherent shots. This process yielded a test set of 200 stories, totaling 800 video shots.

**Metrics:** Following prior works (Wu et al., 2025b; Cheng et al., 2025; Yang et al., 2024b), we evaluate models using automatic metrics, advanced MLLM assessments and user studies. For automatic metrics, we employ CLIP score (Radford et al., 2021) and DreamSim (Fu et al., 2023) to quantify the consistency between generated characters and the reference image at shot-level and story-level. We also leverage GPT-4o and Gemini 2.5 Pro as MLLM-based judges, and conduct user studies to align with human preferences. The evaluation dimensions include Overall Quality [OQ], Character-Reference Consistency [CRC], Multi-Shot Style Consistency [MSC] and Multi-Shot Contextual Consistency [MCC]. More detailed information are in the supplementary.

### 5.2 QUANTITATIVE COMPARISONS

As shown in Table 2, automatic metrics evaluating character-reference alignment demonstrate that AnimeShooterGen outperforms both comparison methods. Notably, despite being trained on sequences of only 3 consecutive shots, AnimeShooterGen generalizes robustly to longer sequences during testing. Table 3 reveals additional advantages through MLLM evaluation and user studies. Beyond achieving superior CRC which also evaluates character-reference alignment, AnimeShooterGen exceeds comparison methods in MSC and MCC. Results underscore its dual strengths: (1) Enhanced reference image alignment. AnimeShooterGen achieves markedly higher character consistency than CogVideo-LoRA which shares the same diffusion architecture, proving that MLLM conditions effectively encode reference image features. (2) Cross-shot visual coherence. The MLLM's memory mechanism retains historical context across shots, enabling high-level semantic alignment to guide the diffusion process in generating stylistically and contextually consistent new shots.

### 5.3 QUALITATIVE COMPARISONS

Figure 4 illustrates the visual outcomes of different methods on multi-shot animation generation task. The IP-Adapter + I2V approach struggles to maintain fidelity to the provided reference images due to weak control over IP-specific features. For instance, in the right-hand example, the generated character exhibits significant discrepancies in hairstyle, clothing, and facial structure compared to the reference image. CogVideo-LoRA also fails to achieve alignment with the reference images. Critically, both comparison methods generate individual shots as independent processes, leading to

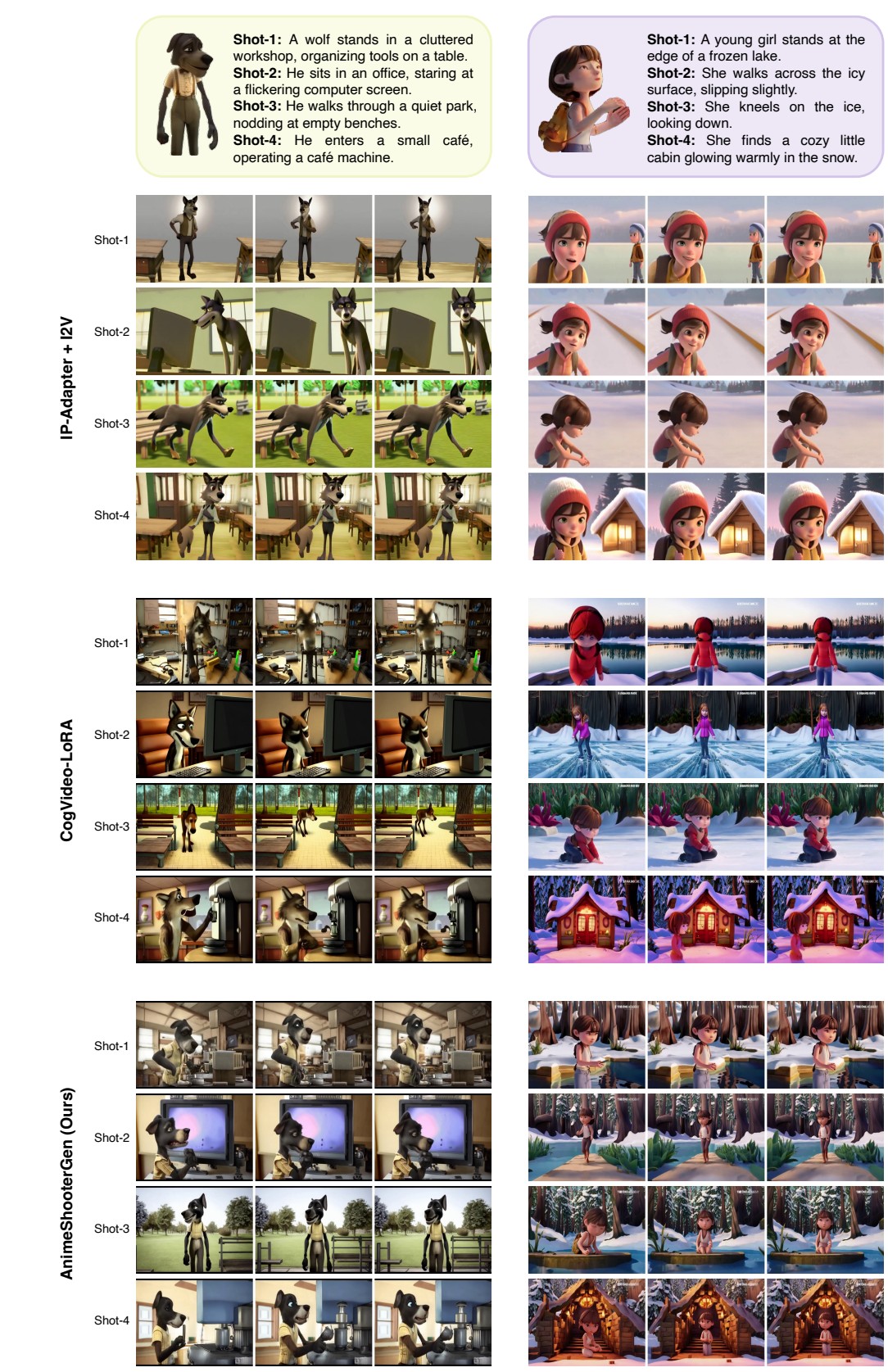

Figure 4: Qualitatively comparisons on multi-shot animation generation. Our method delivers the best visual quality, including character-reference consistency and multi-shot consistency.

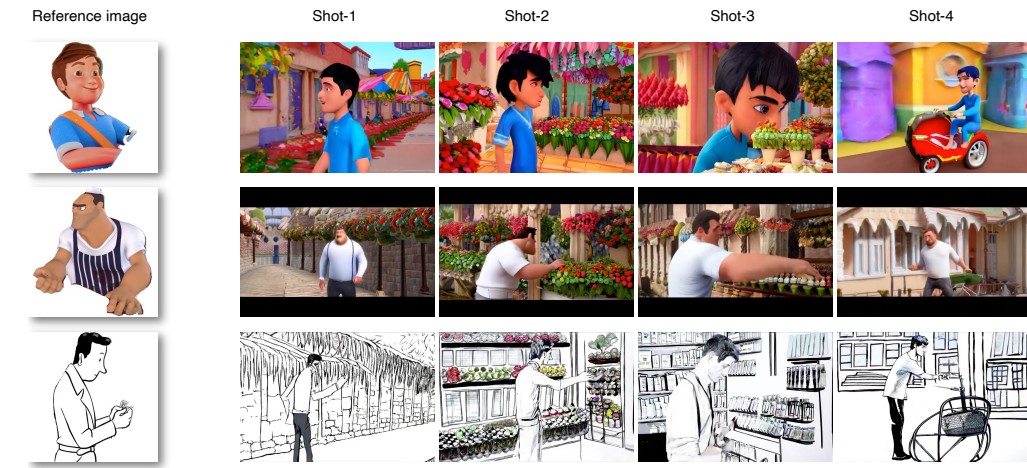

Figure 5: Visualization of using different references in MLLM (**before LoRA Enhancement**). Shared caption: "Shot-1: The man walks down a cobblestone street lined with blooming cherry trees, holding a vintage leather journal under his arm.", "Shot-2: He pauses at a flower shop, steps inside, and begins carefully selecting flowers.", "Shot-3: At the counter, he wraps the bouquet in paper.", "Shot-4: He tucks the flowers into his bicycle basket and pedals away past pastel-colored storefronts."

glaring inconsistencies between shots. In the left-hand example, both comparison methods depict the wolf in its natural animal form in the third shot, with anthropomorphic representations generated in the remaining shots. In contrast, AnimeShooterGen achieves superior reference fidelity, and sustains cross-shot consistency in style and environmental elements, as demonstrated by the invariant morphology of snow-covered trees across multiple shots. It contribute to the autoregressive generation strategy, where previously generated shots directly condition subsequent ones. This mechanism ensures robust style uniformity and contextual coherence throughout the multi-shot sequence.

## 5.4 INVESTIGATING THE IMPACT OF REFERENCE IMAGES

To isolate and understand the direct influence of reference images on the generation process, we omit the LoRA enhancement phase. The model is conditioned on same captions paired with distinct reference images. As illustrated in Figure 5, the reference images inject coarse-grained visual cues into the MLLM condition, influencing both character appearance and artistic style: in rows 1 and 2, the generated characters adopt clothing colors and silhouettes that closely correspond to their respective reference images; the minimalist sketch style in row 3 directly mirrors the reference image's aesthetic. Crucially, even in the absence of LoRA enhancement, the autoregressive nature of our framework maintains strong multi-shot consistency. This observation underscores the inherent capability of the autoregressive architecture to enforce shot-to-shot coherence.

## 6 CONCLUSION AND LIMITATION

This paper introduces AnimeShooter, a comprehensive dataset for reference-guided multi-shot animation generation, featuring comprehensive hierarchical annotations and strong visual consistency across shots. Story-level annotations provide the storyline, key scenes, and main character profiles with reference images, while shot-level annotations decompose the story into consecutive shots, each annotated with scene, characters, and both narrative and descriptive visual captions. We also propose AnimeShooterGen which can generate reference-guided multi-shot animation in an autoregressive manner. Experiments demonstrate that being trained on AnimeShooter's multi-shot annotations promotes cross-shot consistency and adherence to predefined references. Current limitations include the restriction of AnimeShooterGen from open-domain generation due to computational demands, the requirement for test-time fine-tuning to enhance character consistency, and suboptimal audio-visual synchronization resulting from a naive zero-shot audio generation approach. We anticipate AnimeShooter will serve as a valuable resource for future work aimed at developing more robust open-domain models with improved audio-visual alignment and character fidelity.

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

# A DETAILS FOR DATA CURATION

## A.1 MULTI-SHOT CAPTIONING

In Figure 6, we present the detailed prompt designed for multi-shot captioning (Section 3.2) with Gemini-2.0-flash. Initially, Gemini-2.0-flash is prompted to establish a global narrative context by summarizing a succinct, coherent storyline and identifying main characters and main scenes. Following this analysis, it decomposes the whole story into consecutive shots and provides detailed shot-level annotations.

**USER:**
Given a video, your task is to succinctly summarize the storyline and hierarchically decompose it into shots according to the following guidelines:
1. Ensure that the storyline is succinct and coherent.
2. You should identify the main characters and the main scenes (where the story takes place). Provide detailed descriptions of the character's physical appearances, and the environment, style and color of the scenes.
3. The main character list should include only one to three important characters, those who appear repeatedly throughout the storyline. Characters who appear only a few times and are not important to the storyline should be omitted.
4. Decompose this video into non-overlapping shots based on transitions. For each shot, provide timestamps that cover the entire duration without skipping any part of the video. If a shot is longer than 4 seconds, in addition to the full shot duration, select a coherent and complete 4-second segment that best represents the corresponding plot and caption. Make sure all the start time and end time is connected between adjacent shots.
5. For each shot, provide a plot which focus on the action and story plot instead of visual descriptions. For example, "The girl said goodbye to the bear" is preferred over "A girl in red standing in front of a brown bear".
6. For each shot, also provide a detailed descriptive caption of the scene/environment, characters, and their actions or movements, even if these details have been mentioned in previous shots. For example, "By the bank of a river covered with trees, a girl in red stand in front of a brown bear, smiling and waving to him".
7. The story may contain prologue and epilogue, which are usually characterized by a pure background, the appearance of the story title, or a lot of text to indicate creation information. Please mark these meaningless prologue and epilogue that do not advance the plot in your response.

The output format is JSON. Here is an example:

```
{
 "storyline": ...,
 "main characters": [
  {
   "ID": //name of character1,
   "appearance": ...
  },...
 ],
 "main scenes": [
  {
   "ID": //name of scene1,
   "environment": ...
  },...
 ],
 "shots": [
  {
   "start time": "MM:SS",
   "end time": "MM:SS",
   "extra 4-second segment": "MM:SS-MM:SS", //optional, only if the shot is longer than 4 seconds
   "is_prologue_or_epilogue": prologue / epilogue, //optional, only if the shot is prologue or epilogue
   "main characters": [//list IDs of main characters who appear in the current shot]
   "scene": //write ID of the scene of the current shot
   "plot": //write the story plot
   "caption": //write the detailed descriptive caption
  },...
 ]
}
```

<VIDEO>

Figure 6: The prompt used for multi-shot captioning.

## A.2 Reference Image Generation

To address the challenge of ambiguous segmentation in images where multiple characters or non-target individuals are present, we use the prompt template for Sa2VA (Figure 7, top subfigure) that explicitly incorporates character IDs and appearance descriptions from the shot-level character list, followed by a clear specification of the target ID for segmentation. For post-processing, we first apply morphological opening and closing operations with a kernel size of 5 to smooth boundaries and remove noise, and then eliminate masks containing over 15 contours or 5 disconnected components. Finally, we retain only the largest connected region, and discard masks occupying less than 5% or exceeding 90% of the total image area. The prompt shown in the bottom subfigure of Figure 7 is provided to InternVL for the secondary verification.

---

**USER:**
<IMAGE>
This image may contain characters as follows:
1. Girl, A young girl with short dark hair, wearing a pink bow.
2. Boy, A toddler with short dark hair, wearing a yellow overall.
3. Caregiver, An adult male with brown hair, a mustache, and glasses.

Please segment Caregiver.

---

**USER:**
You are tasked with evaluating whether an image can serve as a proper character reference image. Given an image and a character description, analyze the following criteria:

1. Character Completeness:
- The character is segmented from video frames, the segmented character should be complete, not fragmented
- The character's key features and details should be clearly visible

2. Description-Image Consistency:
- Compare the character's appearance in the image with the provided description
- Check if physical attributes (hair color, clothing, body type, etc.) match
- Verify if distinctive features mentioned in the description are present
- Note any inconsistencies or missing elements

3. Text-based Appearance Consistency:
- Ensure the character's appearance is consistent throughout the description
- Check if the description maintains the same physical attributes across different moments
- Reject cases where the description shows inconsistency (e.g., wearing red clothes at one time and blue clothes at another)

4. Pose Evaluation:
- The character should be in a neutral or natural standing pose
- Avoid images with exaggerated expressions or dynamic action poses
- The viewing angle should provide a clear understanding of the character

5. Image Quality:
- The image should be clear and sharp, not blurry
- Details should be easily distinguishable
- Avoid images with motion blur or poor resolution

Remember: A good character reference image should serve as a clear, accurate representation of the character's standard appearance and design.

Based on these criteria, provide:
- Verdict: yes/no on whether the image is suitable as a reference
- Explanation: a brief explanation of your verdict

Here is the character description:
<CHARACTER APPEARANCE>

Here is the image:
<IMAGE>

---

Figure 7: The prompt used for reference image generation. Top subfigure: Segmentation prompt for Sa2VA. Bottom subfigure: Filtering prompt for InternVL.

### A.3 AUDIO ANNOTATION FOR ANIMESHOOTER-AUDIO

This section aims to generate visual-audio annotations by addressing the inherent asynchrony between modalities (e.g., audio transitions lingering beyond visual shot cuts). To ground the model's analysis and maintain consistency with pre-existing holistic information, thus avoiding redundant reference image generation, video segments alongside their story-level annotations are supplied to Gemini-2.5-Pro. Gemini-2.5-Pro is required to firstly exclude video segments with prolonged background music or human speech. Following this, the model executes shot decomposition and shot-level annotation. It performs a joint analysis of visual and auditory cues to detect optimal clip boundaries where both modalities exhibit coherent transitions. For each clip, it generates: (1) two types of visual captions, (2) audio descriptors (categorizing sound types and describing tones), and (3) source attribution (mapping sounds to visual elements). Notably, these clips do not strictly adhere to single visual shot boundaries, as decomposition is determined by the joint consideration of both visual and auditory transitions. For terminological consistency, these audio-visually coherent clips are still referred to as 'shots'. The prompt used for audio annotation is shown in Figure 8.

### A.4 DATASET EXAMPLE

Figure 9 presents an example from AnimeShooter, featuring various scenes such as Luna's room, birthday table, and shoemaker shop, which are further divided into finer shots based on cuts.

## B DETAILS FOR MODEL TRAINING

### B.1 MODEL FRAMEWORK

AnimeShooterGen leverages NVILA-8B (Liu et al., 2024b) as the pretrained MLLM backbone and CogVideo-2B (Hong et al., 2022) as the pretrained video diffusion model. An adapter, crucial for inter-model communication, is implemented using a QFormer (Li et al., 2023) with 12 layers. The length of learnable queries is set to 64. In AnimeShooterGen, for a sequence including $n$ shots, MLLM receives an input formatted as "<Reference><Caption$_1$><LQ><Frame$_1$>...<Caption$_n$><LQ>", where each <Caption$_i$> represents textual guidance for the i-th shot and <LQ> serves as a placeholder of learnable query for contextual feature extraction. During training, the <Frame$_1$> to <Frame$_{n-1}$> tokens are populated with the last frames from their corresponding ground truth video shots, and all $n$ shots contribute jointly to the diffusion loss through backpropagation. At inference time, the model operates autoregressively: it first generates a 1-shot sequence using only the initial reference, then replaces the <Frame$_1$> token with the final frame of the newly generated shot to condition the next iteration, recursively extending the sequence until reaching the target length $n$.

### B.2 IMPLEMENTATION DETAILS

For Condition Alignment, we train the model using video-caption pairs from the large-scale WebVid-10M dataset (Bain et al., 2021). MLLM takes the first frame of a ground-truth video clip, its corresponding caption, and learnable queries as inputs. We minimize the MSE loss between the MLLM's output condition and the T5 features of the caption. During this phase, only the adapter and learnable queries are trainable, optimized with a batch size of 32 for 1.8 M steps.

For Single-Shot Training, we utilize samples from the AnimeShooter dataset containing reference images and descriptive captions for individual shots. Here, the adapter, learnable queries, and LoRA parameters of the MLLM are fine-tuned with a batch size of 24 for 17K steps. Classifier-Free Guidance (CFG) is applied to enhance multimodal control, with independent dropout probabilities of 0.05 for the reference image, caption, or both modalities.

For Multi-Shot Training, we curate 3-shot sequences from the AnimeShooter dataset. The same trainable components (adapter, learnable queries, and MLLM LoRA) are updated with a batch size of 8 for 8K steps. A simplified CFG strategy is adopted, where both the reference image and caption inputs are simultaneously replaced with blank content at a 0.05 probability, eliminating modality-specific dropout.

**USER:**
Analyze the input animation video and perform visual-audio annotation following these steps:
# Video Verification
Ensure the video does not contain extended segments of background music (BGM) or human speech. The audio of this video should feature only sound effects (e.g., footsteps, alarms, animal sounds, environmental sounds).

# Shot Segmentation
Split the video into non-overlapping multi-shot structure using combined visual+audio boundaries where:
- Make sure all the start time and end time is connected between adjacent shots.
- Each shot maintains continuous visual narrative and audio context.
- Transition points must show simultaneous visual+audio changes.
- Each shot lasts more than 3 seconds.
- For each shot, provide start/end timestamps (MM:SS).

# Shot Annotation
Based on the given video-level annotation (main characters, main scenes and storyline), you should create shot-level annotations for each shot:
(1) List main characters present in this shot and the scene of this shot.
(2) The video may contain prologue and epilogue, which are usually characterized by a pure background, the appearance of the story title, or a lot of text to indicate creation information. Please mark these meaningless prologue and epilogue that do not advance the plot in your response.
(3) Visual annotation: For each shot, provide a plot which focus on the action and story plot instead of visual descriptions. For example, "The girl said goodbye to the bear" is preferred over "A girl in red standing in front of a brown bear". For each shot, also provide a detailed descriptive caption of the scene/environment, characters, and their actions or movements, even if these details have been mentioned in previous shots. For example, "By the bank of a river covered with trees, a girl in red stand in front of a brown bear, smiling and waving to him".
(4) Audio annotation: For each shot, provide three description layers for only sound effects. The first is "type", it means sound effect category, e.g., alarm, magical plink, scampering. The second is "sound description", it means technical audio characteristics (pitch, intensity, temporal pattern, etc.). The third is "visual context", it means on-screen source/trigger of sound.

# Output Format
Structure as hierarchical JSON. If the video contains extended segments of background music (BGM) or human speech, return empty JSON object.
json
```
{
  "storyline": ...,
  "main characters": [
    {
      "ID": ...,
      "appearance": ...
    },...
  ],
  "main scenes": [
    {
      "ID": ...,
      "environment": ...
    },...
  ],
  "shots": [
    {
      "start time": "MM:SS",
      "end time": "MM:SS",
      "is_prologue_or_epilogue": prologue / epilogue,  //optional, only if the shot is prologue or epilogue
      "main characters": [//list IDs of main characters who appear in the current shot],
      "scene": ..., //write ID of the scene of the current shot
      "visual annotation": {
        "plot": ...,
        "caption": ...
      },
      "audio annotation": {
        "type": ...,
        "sound description": ...,
        "visual context": ...
      }
    },...
  ]
}
```

# Now, please analyze the following case.
<VIDEO>
<VIDEO LEVEL ANNOTATION>

Figure 8: The prompt used for constructing AnimeShooter-audio.

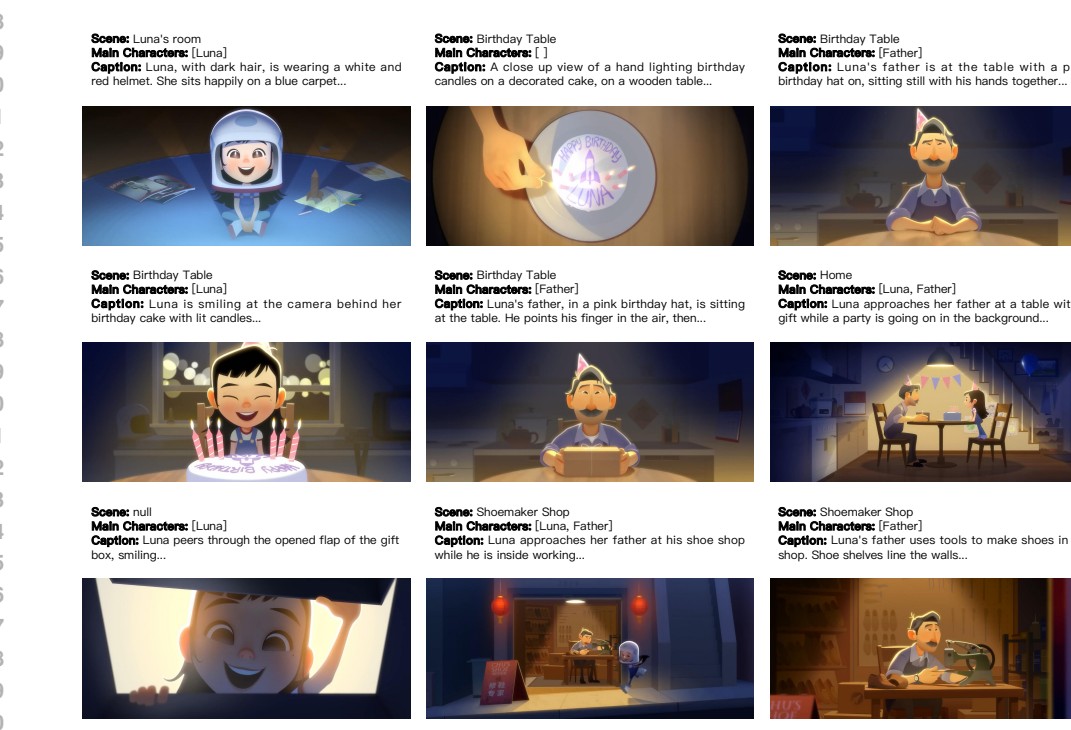

Figure 9: Multi-shot dataset example with different scenes.

For LoRA enhancement, given 5-6 separate video clips from a particular IP, these are randomly sampled and combined to form training sequences of 3 consecutive shots. This targeted finetuning is performed for 1K to 2K steps with batch size of 2.

All training stages are implemented on GPUs with 40G memory. Here are computational costs of different training stages: Condition Alignment for 1400 GPU hours; Single-Shot Training for 900 GPU hours; Multi-Shot Training for 400 GPU hours; LoRA Enhancement for 20 GPU hours. The inference can be implemented on a single GPU with 40G memory with pipeline offloading. The inference time of a story with 4 shots is around 720s.

## C  DETAILS FOR EXPERIMENTS

### C.1  BASELINES

For fair comparison with CogVideo-LoRA, we fine-tune the same pretrained model (CogVideo-2B) on the same IP-specific dataset and iterations as AnimeShooterGen. While AnimeShooterGen is trained in multi-shot mode (batch size=2 per step, with each sample containing 3 shots, totaling 6 shots per step), CogVideo-LoRA supports only single-shot training. To match the computational scale, we set its batch size to 6 (6 shots per step). For the training-free baseline IP-Adapter + I2V, we utilize stable-diffusion-xl-base-1.0 with its IP-Adapter to generate keyframes conditioned on reference images and captions. These keyframes and captions are then fed to the CogVideo-5B I2V model (replacing CogVideo-2B due to its lack of I2V capability) to synthesize video shots.

### C.2  EVALUATION DATASET CONSTRUCTION

We construct a manually annotated evaluation dataset of 20 animated films featuring distinct IPs to support LoRA enhancement for AnimeShooterGen and finetuning for CogVideo-LoRA. For each IP, alongside a reference image, we curate 5–6 short video clips (each lasting several seconds) exclusively depicting the main character, ensuring maximal diversity in actions and scenes, as shown in Figure 10. To evaluate multi-shot generation performance, we employ DeepSeek with prompt shown in Figure 11 to generate 10 unique narrative prompts per IP.

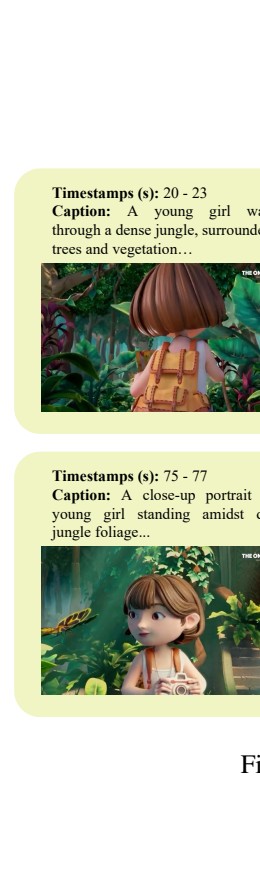
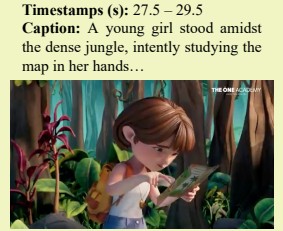
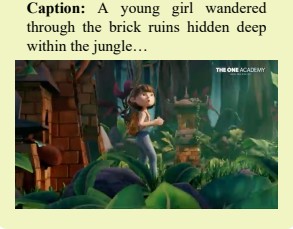
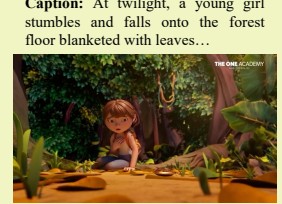
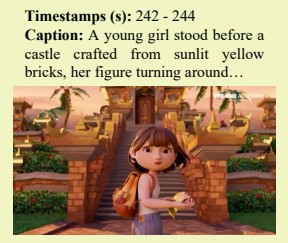

**Timestamps (s):** 20 - 23
**Caption:** A young girl walked through a dense jungle, surrounded by trees and vegetation…

**Timestamps (s):** 27.5 – 29.5
**Caption:** A young girl stood amidst the dense jungle, intently studying the map in her hands…

**Timestamps (s):** 48 – 52
**Caption:** A young girl wandered through the brick ruins hidden deep within the jungle…

**Timestamps (s):** 75 - 77
**Caption:** A close-up portrait of a young girl standing amidst dense jungle foliage...

**Timestamps (s):** 140 - 149
**Caption:** At twilight, a young girl stumbles and falls onto the forest floor blanketed with leaves…

**Timestamps (s):** 242 - 244
**Caption:** A young girl stood before a castle crafted from sunlit yellow bricks, her figure turning around…

Figure 10: Example of IP-specific dataset for evaluation.

**USER:**
Generate 10 prompts for evaluation cases. Each prompt must contain 4 consecutive scenes with simple captions following these rules:
1. Storytelling format with scene continuity but distinct background changes and different events.
   - Fixed character reference (it will be provided)
   - Only one character in all scenes
2. Keep captions simple.
   - Use basic verbs and no complex interactions.
   - Use simple language without complicated adjectives or descriptive phrases.
   - Without small objects or detailed props.
3. Follow this exact JSON format:
```
{
 "prompts": [
   [
     "scene1 description",
     "scene2 description",
     "scene3 description",
     "scene4 description"
   ],
   // 9 more prompts
  ]
}
```

Now, please generate prompts for a young girl.

Figure 11: The prompt used for constructing evaluation dataset.

### C.3 AUTOMATIC METRICS

We employ CLIP score and DreamSim to quantify the visual similarity between generated characters and the reference image. We uniformly sample 5 frames from each shot and compute the average similarity score as the shot-level metric. To assess story-level consistency across 4 shots, we introduce two metrics: (1) Mean Similarity (Mean), the arithmetic mean of the 4 shot-level similarity scores. (2) Penalized Harmonic Mean Similarity (HarMeanP). Recognizing that even a single poorly generated shot can disrupt viewer immersion, this metric penalizes the worst-performing shot. This metric first calculates the harmonic mean of all 4 shots' similarity scores (chosen for its sensitivity to extremely low values), then multiplies this result by the lowest similarity score as an additional penalty term.

### C.4 MLLM ASSESSMENT

We leverage GPT-4o and Gemini 2.5 Pro as MLLM-based judges. To mitigate ordering bias in evaluation, we employ the prompt template shown in Figure 12 and change the presentation order across three independent evaluation rounds. The final results reported in Section 5.2 represent the averaged metrics from these three trials, ensuring robustness against positional preferences. MLLM uses 1-10 scoring (1=worst, 10=best) with one-point increments. Evaluation dimensions including:

- Overall quality [OQ]. A holistic assessment considering aesthetic appeal, image quality, visual consistency and so on.
- Character-Reference Consistency [CRC]. Visual fidelity of the generated character to the provided reference image.
- Multi-Shot Style Consistency [MSC]. Coherence of artistic style, color palette, and texture across all shots within a story.
- Multi-Shot Contextual Consistency [MCC]. Continuity of the narrative and context across shots, e.g., ensuring the character maintains a consistent appearance appropriate to the unfolding story.

### C.5 HUMAN EVALUATION

For the human evaluation, we recruit 10 participants who hold at least a bachelor's degree and have prior experience in image or video generation. A total of 15 stories with 60 shots are presented to the participants. The evaluation metrics align with those defined in Section C.4. Participants are instructed to score each dimension (1: lowest, 5: highest) based on the reference image, corresponding captions, and multi-shot videos generated by the three models. The final performance of each model is calculated as the average scores across all responses.

### C.6 MODEL PERFORMANCE ON LONGER STORYLINES

To investigate the performance of AnimeShooterGen on longer sequences, we utilize DeepSeek to generate longer stories comprising 15 shots. To manage computational demands, we employ a sliding window inference strategy for shots beyond position 4: each shot $i$ is generated using the previous three shots ($i-3, i-2, i-1$) as its context. These 15-shot sequences are divided into three segments (shots 1-5, 6-10, and 11-15). We present these segments to GPT and Gemini in three different orders, minimizing potential position bias. We also calculate CLIP and DreamSim metrics. Results shown in Table 4 reveal that the model maintains relatively robust performance on extended sequences, but still has slight performance degradation due to error accumulation. We show qualitative results in Figure 13. While quantitative results indicate a slight decline in character similarity over extended sequences, this degradation is hardly noticeable in qualitative assessments, further demonstrating the robustness of the method in temporal extension. However, we also observe other challenges arising from long sequences, such as increased distortion in backgrounds (first row) and characters (second row), as well as a gradual drift in color tone (last row).

### C.7 ADDITIONAL QUALITATIVE RESULTS

We present additional qualitative results in Figure 14 and Figure 15. Both comparison methods exhibit limitations in preserving character consistency with the reference image and cross-shot coherence. As

```
USER:
Please act as an impartial judge and evaluate three AI-generated multi-shot video samples (4 shots each) based on the
following inputs and requirements.

# Input
1. Character reference image (provided first)
2. Text captions for each of the 4 shots (provided second)
3. Three models' outputs - Each model's output is a 4-row grid, where:
Each row corresponds to one shot (Shot 1 to Shot 4 from top to bottom).
Within each row, 3 sampled frames are placed horizontally (left to right).

# Evaluation Dimensions
Use 1-10 scoring (1=worst, 10=best) with one-point increments allowed:

1. Overall Quality <OQ>
Conduct a comprehensive assessment of the overall quality for each model-generated sample, evaluating multiple dimensions
such as visual consistency, aesthetic appeal, image quality, and other relevant factors.

2. Character-Reference Consistency <CRC>
Evaluate visual fidelity and consistency between generated characters and the provided reference image.

3. Multi-Shot Style Consistency <MSC>
Assess style, color palette and texture coherence across all shots.

4. Multi-Shot Contextual Consistency <MCC>
Verify contextual continuity. E.g., evaluate whether the character maintain a stable appearance, behavior, and role throughout
shots.

# Requirements & Output Format
Avoid any bias, ensure that the order of presentation does not affect your decision. Do not favor certain agent names. Reflect
the score differences as much as possible.

Return JSON format with detailed feedback:
{
  "Model_A": {
    "scores": {
      "OQ": 4,
      "CRC": 7,
      "MSC": 3,
      "MCC": 1
    },
    "feedback": {
      "strengths": ["Consistent eye detail", "Good lighting continuity"],
      "weaknesses": ["Hand proportions vary in Shot 3", "Background mismatch in Shot 4"]
    }
  },
  "Model_B": {...},
  "Model_C": {...},
  "comparative_analysis": "Model A excels in character consistency and performs best..."
}
```

Figure 12: The prompt used for MLLM assessment.

Table 4: Model Performance on Longer Storylines.

| Shots | CLIP ↑ | DreamSim ↓ | OQ ↑ | | CRC ↑ | | MSC ↑ | | MCC ↑ | |
|---|---|---|---|---|---|---|---|---|---|---|
| | | | GPT | Gem. | GPT | Gem. | GPT | Gem. | GPT | Gem. |
| Shot 1-5 | 0.7989 | 0.3637 | 7.19 | 6.99 | 7.94 | 7.59 | 6.84 | 7.34 | 6.45 | 6.37 |
| Shot 6-10 | 0.7945 | 0.3778 | 6.83 | 6.46 | 7.67 | 7.26 | 6.52 | 7.21 | 6.01 | 5.51 |
| Shot 11-15 | 0.7961 | 0.3746 | 6.96 | 6.52 | 7.76 | 7.08 | 6.64 | 7.11 | 6.18 | 5.61 |

illustrated in the right panel case of Figure 15, the IP-Adapter+I2V framework erroneously transforms
the flashlight-equipped helmet into a color-mismatched hat. In the left panel case, the first shot
generated by CogVideo-LoRA demonstrates a visually discordant art style. In contrast, our proposed
AnimeShooterGen achieves superior preservation of character identity, color palette continuity, and
stylistic consistency across generated sequences.

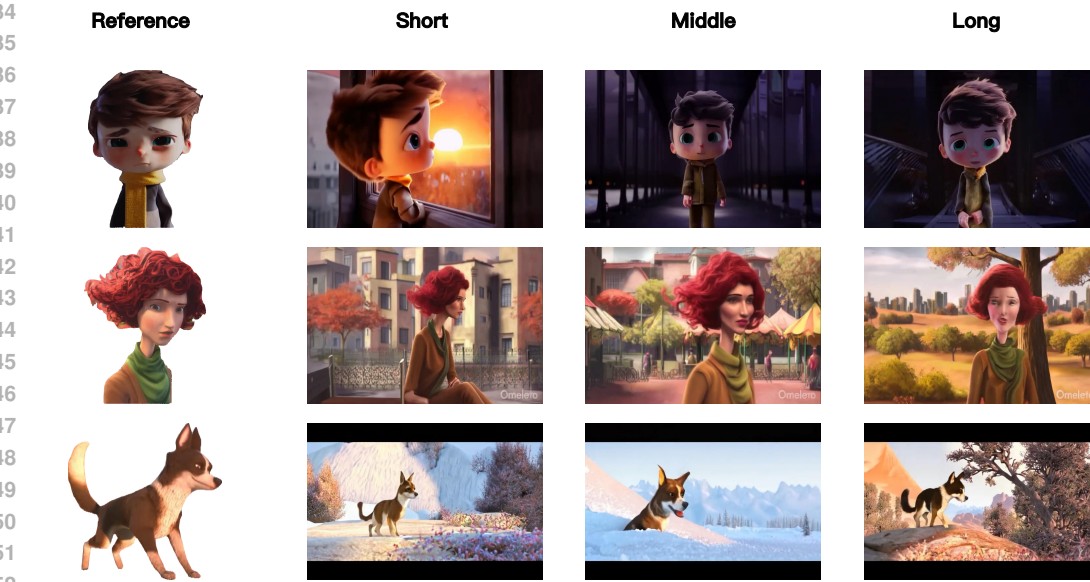

Figure 13: Qualitative results for longer storylines.

Table 5: Results of evaluation on VBench.

| Model | Subject Cons. | Bg Cons. | Motion Smth. | Dyn. Degree | Aesthetic | Imaging |
|---|---|---|---|---|---|---|
| IP-Adapter + I2V | 0.9341 | **0.9543** | 0.9873 | 0.6500 | 0.5959 | 0.6073 |
| Cogvideo-LoRA | 0.9330 | 0.9539 | 0.9904 | 0.7025 | 0.6171 | **0.6524** |
| AnimeShooterGen | **0.9366** | 0.9517 | **0.9921** | **0.7124** | **0.6255** | 0.6512 |

## C.8 VIDEO QUALITY

Table 5 presents the evaluation results on VBench (Huang et al., 2024b). Although IP-Adapter + I2V employs the more powerful CogVideo-5B model, it does not demonstrate superior video quality, likely due to the domain gap between real-world and animation. In contrast, CogVideo-LoRA and AnimeShooterGen which are trained on animation dataset achieve comparable or even better quality with only a 2B model, and AnimeShooterGen achieves the best performance across most metrics.

## C.9 ABLATION STUDIES

Table 6 presents the results of ablation study, which includes the following configurations: (1) w/o MLLM condition, where the MLLM condition is replaced with the original text condition (i.e., CogVideo-LoRA); (2) w/o LoRA Enhancement, where no LoRA fine-tuning based on a specific IP, and zero-shot capability is tested directly; and (3) w/o Reference Image, where the reference image in the MLLM is replaced with a blank image. The results further confirm that after training on the AnimeShooter dataset, the MLLM is able to capture high-level semantic consistency. Replacing it with the diffusion model's original text condition or using a blank image significantly degrades model performance. Moreover, the results highlight the importance of the detail alignment achieved during the LoRA Enhancement stage, and the high-level semantic consistency provided by the MLLM cannot be fully realized at the pixel level without it.

## C.10 IMPACT OF DIFFERENT VISUAL CONTEXT FRAMES

This section examines how the choice of different frames as the visual context influences model performance. We present in Table 7 the results under different configurations: using the first frame (first), the middle frame (mid), the last frame (last), these three frames simultaneously (multi), and the pooled feature of them (pool). The results demonstrate that the choice of frame has no notable impact on model performance. Moreover, incorporating additional frames or using feature pooling

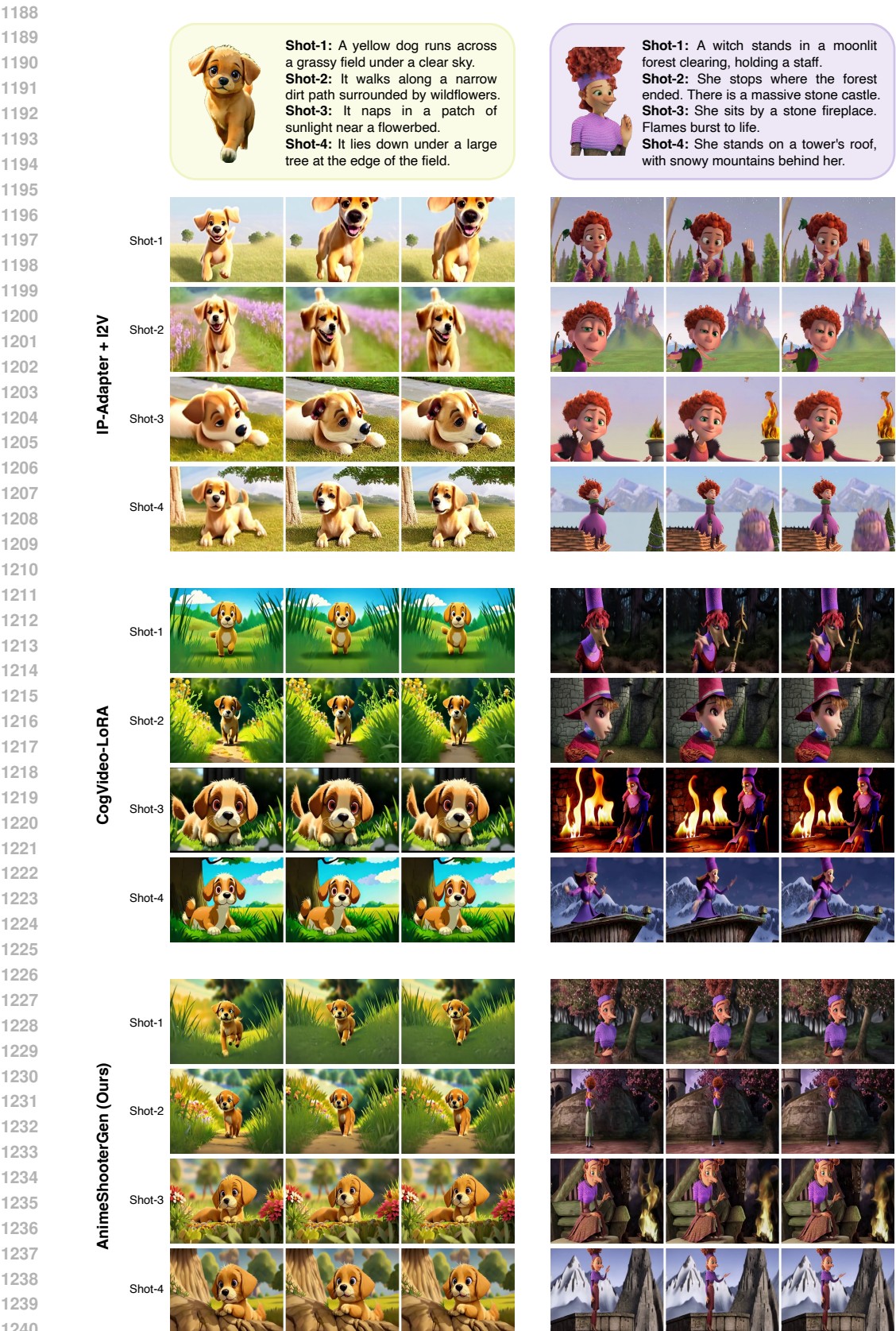

Figure 14: Additional qualitative results.

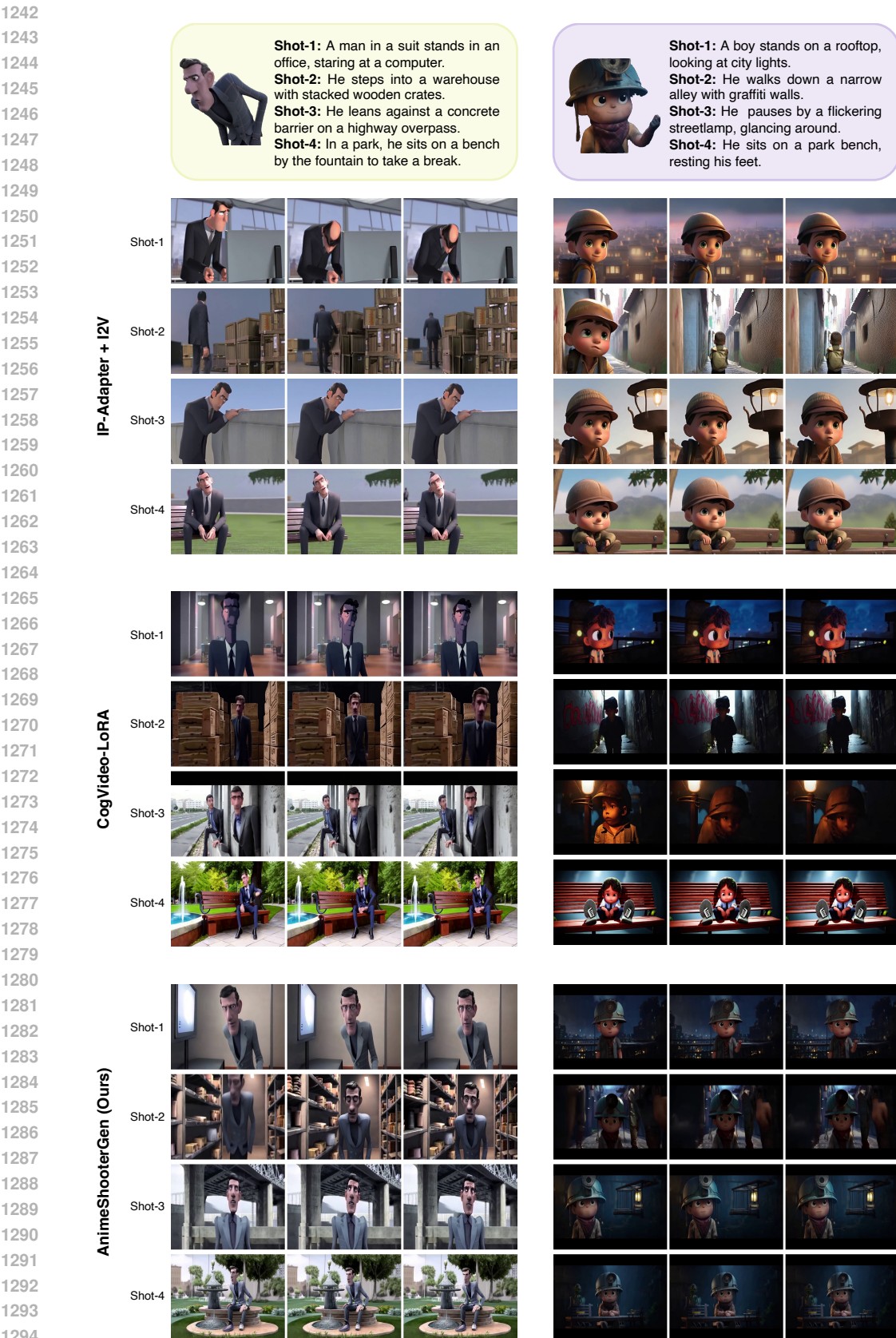

Figure 15: Additional qualitative results.

Table 6: Results of ablation studies.

| Model | Metric | Shot-level | | | | Story-level | |
|---|---|---|---|---|---|---|---|
| | | Shot-1 | Shot-2 | Shot-3 | Shot-4 | Mean | HarMeanP |
| w/o MLLM condition | | 0.7297 | 0.7200 | 0.7417 | 0.7413 | 0.7332 | 0.5028 |
| w/o LoRA Enhancement | CLIP ↑ | 0.7628 | 0.7430 | 0.7376 | 0.7362 | 0.7449 | 0.5280 |
| w/o Reference Image | | 0.7065 | 0.7167 | 0.7208 | 0.7264 | 0.7176 | 0.4934 |
| AnimeShooterGen | | **0.8022** | **0.7949** | **0.7970** | **0.7986** | **0.7982** | **0.6121** |
| w/o MLLM condition | | 0.4777 | 0.5060 | 0.4842 | 0.4864 | 0.4886 | 0.7759 |
| w/o LoRA Enhancement | DreamSim ↓ | 0.4773 | 0.5198 | 0.5178 | 0.5323 | 0.5118 | 0.7940 |
| w/o Reference Image | | 0.5363 | 0.5348 | 0.5213 | 0.5131 | 0.5264 | 0.7926 |
| AnimeShooterGen | | **0.3484** | **0.3820** | **0.3799** | **0.3764** | **0.3717** | **0.6413** |

Table 7: Impact of different visual context frames.

| Visual context frame | Metric | Shot-level | | | | Story-level | |
|---|---|---|---|---|---|---|---|
| | | Shot-1 | Shot-2 | Shot-3 | Shot-4 | Mean | HarMeanP |
| first | | 0.8152 | 0.8017 | 0.8068 | 0.8115 | 0.8088 | 0.6286 |
| mid | | 0.8108 | 0.8033 | 0.7992 | 0.8113 | 0.8062 | 0.6238 |
| last | CLIP ↑ | 0.8168 | 0.8055 | 0.8116 | 0.8094 | 0.8108 | 0.6323 |
| multi | | 0.8103 | 0.7995 | 0.8059 | 0.8031 | 0.8047 | 0.6223 |
| pool | | 0.8166 | 0.8007 | 0.8007 | 0.8025 | 0.8051 | 0.6205 |
| first | | 0.3318 | 0.3697 | 0.3639 | 0.3591 | 0.3561 | 0.6205 |
| mid | | 0.3304 | 0.3676 | 0.3692 | 0.3599 | 0.3568 | 0.6222 |
| last | DreamSim ↓ | 0.3325 | 0.3714 | 0.3625 | 0.3590 | 0.3563 | 0.6226 |
| multi | | 0.3344 | 0.3758 | 0.3740 | 0.3750 | 0.3648 | 0.6334 |
| pool | | 0.3355 | 0.3809 | 0.3701 | 0.3652 | 0.3629 | 0.6307 |

does not lead to performance improvement. Considering computational efficiency, a single frame suffices as the visual context, as it already contains sufficient visual information.

## D INTEGRATION OF AUDIO GENERATION CAPABILITIES FOR ANIMESHOOTERGEN

To further augment the immersive quality, we integrate AnimeShooterGen with zero-shot Text-to-Audio (TTA) generation using TangoFlux (Hung et al., 2024). The workflow involves processing the video captions and keyframes with GPT-4o to generate descriptive audio captions with the prompt shown in Figure 16. These audio captions subsequently guide TangoFlux in synthesizing audio tracks, which are then merged with the video sequences. However, results reveal substantial limitations in current simplistic zero-shot audio generation paradigms. Primarily, the decoupled generation processes for visual and auditory modalities result in inherent inter-modality synchronization failures. For example, footstep sounds lag behind walking animations, or character facial expressions mismatch with voice. Furthermore, constrained by existing text-to-audio models' performance, environmental sound effects such as gentle breezes, engine roars, and mechanical hums fail to achieve sufficient perceptual distinctiveness, thereby compromising the immersive experience. We propose an audio-annotated subset named AnimeShooter-audio, hoping to facilitate and encourage future research into the development of more sophisticated audio-visual co-generation models capable of achieving tighter synchronization and semantic coherence.

## E POTENTIAL SOCIAL IMPACTS

While generation models democratize content creation, they risk enabling malicious applications such as generating deepfakes for disinformation and producing harmful content. To address these risks, the research community and policymakers must adopt proactive safeguards. Technical countermeasures

```
USER:
Analyze the three provided animation frames and their corresponding text caption. Generate a descriptive audio prompt for a
text-to-audio model to create corresponding ambient sounds.

# Requirements:
- Prioritize sounds logically implied by the visuals and caption, e.g., dog barks if a dog is shown.
- Specify the source of each sound explicitly, e.g., whirring of a coffee machine, rustling of leaves.
- Sound effects may include: ambience (e.g., wind, city noise), action sounds (e.g., footsteps, door creaks)...
- Avoid abstract metaphors, focus on concrete sounds a model can generate.

# Example Output:
Only one phrase representing one type of sound. Directly output the audio prompt, no other text.
For example:
"a cow is mooning"
"A stream flows"

# Generate the audio prompt now.
<CAPTION>
<FRAMES>
```

Figure 16: The prompt used for generating descriptive audio captions.

include embedding watermarks in generated videos for provenance tracking, deploying AI-driven detectors to flag synthetic content, and implementing strict filters to block unethical prompts.

In light of ethical considerations, we take deliberate measures to ensure the responsible and legal use of data. Our dataset does not involve the distribution of any copyrighted video content. We only release binary masks and text annotations. All copyrights for videos remain with the original creators and YouTube. We provide video IDs to allow access to the source material, and fully respect the creators' ownership and control over their work. By focusing on the animation domain, our research also circumvents privacy concerns associated with real-world videos.

## F    LICENSE

The AnimeShooter dataset is released under the CC BY-NC 4.0 License and the data is collected from publicly accessible sources. The released dataset includes annotated scripts, reference image masks, and corresponding video IDs, while the original source videos must be obtained independently from YouTube using the provided IDs. AnimeShooterGen is built upon two pretrained models: NVILA and CogVideo. We release the code and model weights of AnimeShooterGen under the Apache 2.0 License, and provide a copy of original licenses of NVILA and CogVideo in our GitHub repository.

## G    THE USAGE OF LARGE LANGUAGE MODELS

We utilize Large Language Models solely for language editing and proofreading of the manuscript. All research, including ideation, literature review, and analysis, as well as the initial drafting, is conducted by the authors without the assistance of AI.

