# OpenReview forum: "AnimeShooter: A Multi-Shot Animation Dataset for Reference-Guided Video Generation"
_ICLR.cc/2026/Conference — Submitted to ICLR 2026_

### Official Review · Reviewer_Kftk · 2025-10-21

**Soundness:** 3
**Presentation:** 2
**Contribution:** 3
**Rating:** 6
**Confidence:** 4

**Summary:**

This paper introduces AnimeShooter, a new dataset designed to address the task of reference-guided, multi-shot animation generation. The AnimeShooter dataset is constructed through an automated pipeline that collects animation videos from YouTube, uses Gemini to generate hierarchical story scripts including story-level and shot-level annotations, and leverages models like Sa2VA and InternVL to extract and filter high-quality character reference images.
To validate the dataset's effectiveness, the authors propose a baseline model, AnimeShooterGen, which uses MLLM and video diffusion models to generate subsequent shot according to the reference and context. Experimental results demonstrate the effectiveness of the AnimeShooter dataset and AnimeshooterGen.

**Strengths:**

- The primary contribution is a large-scale dataset for the animation domain that addresses a clear and important research gap in multi-shot, reference-guided video generation. Its scale and rich hierarchical annotations make it a valuable resource for the community.
- The authors employ a highly automated pipeline to process data. And the data is carefully designed.
- To demonstrate the dataset's effectiveness and practical utility, the authors also introduce AnimeShooterGen. This model serves as a strong baseline and effectively validates that the proposed dataset can be used to train robust models for this complex task.

**Weaknesses:**

- The baseline model's strong performance heavily relies on the fourth stage, "LoRA Enhancement," which is essentially test-time finetuning on a few video clips of a specific IP. This makes the model more of a few-shot IP customization method rather than a general reference-guided generator. Can the authors provide a quantitative ablation study comparing AnimeShooterGen's performance on all metrics with and without the LoRA Enhancement stage?
- How does the model perform in a zero-shot setting? That is, given a reference image for an IP that was unseen during all training stages (including LoRA enhancement), how does AnimeShooterGen compare to baselines like IP-Adapter?
- Why was the decision made to use only the last frame of the previous shot as visual context? Were other representations (e.g., first and last frames, multiple frames, pooled video features) explored?
- The dataset provides both a narrative caption and a descriptive caption for each shot. How were these two caption types used when training the baseline model? Were they concatenated, or was only one type (e.g., the descriptive caption) used?

**Questions:**

Please See Weaknesses

**Details Of Ethics Concerns:**

As the authors correctly note in the paper, while generation models democratize content creation, they also risk enabling malicious applications such as generating deepfakes for disinformation and producing harmful content.

---

> ### Author Response · Authors · 2025-11-22
> **Response to reviewer Kftk (W1)**
>
> **W1. The baseline model's strong performance heavily relies on the fourth stage, "LoRA Enhancement," which is essentially test-time finetuning on a few video clips of a specific IP. This makes the model more of a few-shot IP customization method rather than a general reference-guided generator. Can the authors provide a quantitative ablation study comparing AnimeShooterGen's performance on all metrics with and without the LoRA Enhancement stage?**
>
> **Response:** We do not recommend directly using models that haven't been trained with the LoRA enhancement stage, as the MLLM can only provide high-level semantic consistency without detailed alignment. Moreover, the diffusion model remains frozen throughout all preceding stages, and replacing its original text condition with the MLLM's input inevitably introduces bias. We add experiments to show this quantitative results, including the automatic metrics and MLLM assessments. For MLLM assessments, we present comparative results to both Gemini and GPT for model variants with and without LoRA enhancement. And here are the results:
>
> | Automatic Metrics (CLIP$\uparrow$) | Shot1 | Shot2 | Shot3 | Shot4 | Mean | HarMeanP |
> | ----- | ----- | ----- | ----- | ----- | ----- | ----- |
> | w/o LoRA | 0.7628 | 0.7430 | 0.7376 | 0.7362 | 0.7449 | 0.5280 |
> | w LoRA   | **0.8022** | **0.7949** | **0.7970** | **0.7986** | **0.7982** | **0.6121** |
>
> | Automatic Metrics (DreamSim$\downarrow$) | Shot1 | Shot2 | Shot3 | Shot4 | Mean | HarMeanP |
> | ----- | ----- | ----- | ----- | ----- | ----- | ----- |
> | w/o LoRA | 0.4773 | 0.5198 | 0.5178 | 0.5323 | 0.5118 | 0.7940 |
> | w LoRA   | **0.3484** | **0.3820** | **0.3799** | **0.3764** | **0.3717** | **0.6413** |
>
>
> | MLLM Metrics by GPT | OQ$\uparrow$ | CRC$\uparrow$ | MSC$\uparrow$ | MCC$\uparrow$ |
> | -------- | -------- | -------- | -------- | -------- |
> | w/o LoRA | 6.57     | 5.99     | 6.40     | 5.67     |
> | w LoRA   | **7.74**     | **8.51**     | **7.67**     | **7.24**     |
>
> | MLLM Metrics by Gemini | OQ$\uparrow$ | CRC$\uparrow$ | MSC$\uparrow$ | MCC$\uparrow$ |
> | -------- | -------- | -------- | -------- | -------- |
> | w/o LoRA | 4.33     | 2.68     | 5.02     | 4.35     |
> | w LoRA   | **7.43**     | **8.43**     | **8.25**     | **7.23**     |
>
> We acknowledge that achieving general customization without requiring test-time fine-tuning is a more desirable objective, but our design choice was primarily driven by computational constraints and the state of the field at the time of development. There were no open-source video foundation models possessing inherent open-domain customization capabilities that we can utilize to post-train. And our extensive preliminary experiments corroborated by existing works on single-shot customization, revealed that endowing a standard video model (like CogVideo) with general zero-shot customization capabilities is non-trivial and computationally prohibitive. Consequently, inspired by prior works such as DreamRunner[1] and MovieAgent[2], we adopted test-time fine-tuning via LoRA as a pragmatic and efficient alternative to ensure high fidelity.
>
> We also want to emphasize that the primary contribution of this paper is the AnimeShooter dataset, with AnimeShooterGen serving as a baseline to validate its utility. Prior to this work, large-scale public datasets supporting reference-guided customization were virtually non-existent, forcing researchers to rely on internal datasets. By releasing AnimeShooter, we aim to provide the community with the necessary data foundation to train more robust, open-domain customization models (e.g., a "Video IP-Adapter") that can eventually eliminate the need for test-time fine-tuning.
>
> > [1] DREAMRUNNER: Fine-Grained Storytelling Video Generation with Retrieval-Augmented Motion Adaptation
>
> > [2] Automated Movie Generation via Multi-Agent CoT Planning

---

> ### Author Response · Authors · 2025-11-22
> **Response to reviewer Kftk (W2, W3, W4)**
>
> **W2. How does the model perform in a zero-shot setting? That is, given a reference image for an IP that was unseen during all training stages (including LoRA enhancement), how does AnimeShooterGen compare to baselines like IP-Adapter?**
>
> **Response:** Table 2 in manuscript presents the results of IP-Adapter + I2V, along with the zero-shot results of our model (w/o LoRA) discussed in the response to weakness 1, which are compiled and shown together in the table below. As emphasized in the response to weakness 1, AnimeShooterGen currently cannot serve as a "video IP-Adapter". This is because the MLLM in AnimeShooterGen only provides high-level semantic information and lacks low-level visual details. In contrast, general customization models typically inject CLIP or VAE features directly into the diffusion via cross-attention or feature concatenation. Relying solely on the high-level condition from the MLLM is insufficient to accurately reconstruct appearance details.
>
> | Automatic Metrics (CLIP$\uparrow$) | Shot1 | Shot2 | Shot3 | Shot4 | Mean | HarMeanP |
> | ----- | ----- | ----- | ----- | ----- | ----- | ----- |
> | Zero-shot (w/o LoRA) | 0.7628 | 0.7430 | 0.7376 | 0.7362 | 0.7449 | 0.5280 |
> | IP-Adapter + I2V | 0.8004 | 0.7814 | 0.7891 | 0.7947 | 0.7914 | 0.5901 |
>
> | Automatic Metrics (DreamSim$\downarrow$) | Shot1 | Shot2 | Shot3 | Shot4 | Mean | HarMeanP |
> | ----- | ----- | ----- | ----- | ----- | ----- | ----- |
> | Zero-shot (w/o LoRA) | 0.4773 | 0.5198 | 0.5178 | 0.5323 | 0.5118 | 0.7940 |
> | IP-Adapter + I2V | 0.3679 | 0.4169 | 0.4047 | 0.3870 | 0.3941 | 0.6818 |
>
> **W3. Why was the decision made to use only the last frame of the previous shot as visual context? Were other representations (e.g., first and last frames, multiple frames, pooled video features) explored?**
>
> **Response:** The selection of the last frame as the visual context in AnimeShooterGen follows a conventional practice adopted in existing work on long video generation. Many studies focusing on smooth shot transitions tend to condition the generation on the final frame of the preceding shot. As smooth transition is not the objective in our framework, alternative frames could also be feasibly employed.
>
> We have supplemented Appendix C.10 (Impact of Different Visual Context Frames) to examine how the choice of different frames as the visual context influences model performance. We present in following table (Table 7 in manuscript) the results under different configurations: using the first frame (first), the middle frame (mid), the last frame (last), these three frames simultaneously (multi), and the pooled feature of them (pool). The results demonstrate that the choice of frame has no notable impact on model performance. Moreover, incorporating additional frames or using feature pooling does not lead to performance improvement. Considering computational efficiency, a single frame suffices as the visual context, as it already contains sufficient visual information.
>
> | Automatic Metrics (CLIP$\uparrow$) | Shot1 | Shot2 | Shot3 | Shot4 | Mean | HarMeanP |
> | ----- | ----- | ----- | ----- | ----- | ----- | ----- |
> | first | 0.8152 | 0.8017 | 0.8068 | 0.8115 | 0.8088 | 0.6286 |
> | mid   | 0.8108 | 0.8033 | 0.7992 | 0.8113 | 0.8062 | 0.6238 |
> | last  | 0.8168 | 0.8055 | 0.8116 | 0.8094 | 0.8108 | 0.6323 |
> | multi | 0.8103 | 0.7995 | 0.8059 | 0.8031 | 0.8047 | 0.6223 |
> | pool  | 0.8166 | 0.8007 | 0.8007 | 0.8025 | 0.8051 | 0.6205 |
>
> | Automatic Metrics (DreamSim$\downarrow$) | Shot1 | Shot2 | Shot3 | Shot4 | Mean | HarMeanP |
> | ----- | ----- | ----- | ----- | ----- | ----- | ----- |
> | first | 0.3318 | 0.3697 | 0.3639 | 0.3591 | 0.3561 | 0.6205 |
> | mid   | 0.3304 | 0.3676 | 0.3692 | 0.3599 | 0.3568 | 0.6222 |
> | last  | 0.3325 | 0.3714 | 0.3625 | 0.3590 | 0.3563 | 0.6226 |
> | multi | 0.3344 | 0.3758 | 0.3740 | 0.3750 | 0.3648 | 0.6334 |
> | pool  | 0.3355 | 0.3809 | 0.3701 | 0.3652 | 0.3629 | 0.6307 |
>
>
> **W4. The dataset provides both a narrative caption and a descriptive caption for each shot. How were these two caption types used when training the baseline model? Were they concatenated, or was only one type (e.g., the descriptive caption) used?**
>
> **Response:** During the training of the baseline models, we exclusively utilized the descriptive captions and did not concatenate them with narrative captions. This decision was driven by the fact that descriptive captions provide more concrete and fine-grained visual details, which alleviates the model's learning burden by offering clearer guidance for visual synthesis. However, we posit that a narrative storytelling tone is a unique characteristic that distinguishes animation generation from general text-to-video tasks. So we released both types to ensure the dataset's comprehensiveness, aiming to accommodate potential future research requirements.

---

> > ### Comment · Reviewer_Kftk · 2025-11-24
> >
> > I appreciate the authors' efforts in the rebuttal. I believe my current score is a fair reflection of the paper and I decide to maintain my score.

---

### Official Review · Reviewer_pjXJ · 2025-11-01

**Soundness:** 3
**Presentation:** 3
**Contribution:** 3
**Rating:** 6
**Confidence:** 3

**Summary:**

The paper introduces AnimeShooter, a large animation‑focused dataset aimed at reference‑guided, multi‑shot video generation. Each roughly one‑minute “story” includes (i) story‑level annotations (storyline, 1–3 main characters with reference images, and main scenes) and (ii) shot‑level annotations (ordered shots with scene, characters, and both narrative and descriptive captions). A smaller AnimeShooter‑audio subset adds synchronized shot‑level audio descriptions and sources. To demonstrate utility, the authors propose AnimeShooterGen, an autoregressive pipeline that conditions a video diffusion model on: (a) a user reference image, (b) the last frames of previously generated shots, and (c) the shot text. An MLLM backbone produces a conditioning embedding (via a Q‑Former adapter) for a DiT‑based video generator; LoRA layers enable light test‑time adaptation. On a custom multi‑IP evaluation set, AnimeShooterGen outperforms existing baselines on CLIP similarity and DreamSim (shot‑ and story‑level), and in MLLM and user studies.

**Strengths:**

1. The paper isolates a genuinely under‑served setting: reference‑guided multi‑shot animation generation with cross‑shot character/style consistency, rather than single‑shot real‑world videos or global captions.
2. The combination of story‑level elements (storyline, scenes, character cards + reference images) and fine‑grained shot‑level captions is valuable for autoregressive modeling and evaluation.
3. Conditioning on a reference image + prior last frames + text through an MLLM + Q‑Former adapter is technically sound and well motivated.
4. Good empirical results on the proposed evaluation benchmark compared with baselines.

**Weaknesses:**

1. The compared baselines are relatively weaker baselines (e.g., IP‑Adapter+I2V and CogVideo‑LoRA). Stronger baseline models with MLLM might also be considered.
2. The evaluation metrics with CLIP and DreamSim cannot capture some video quality aspects like motion smoothness. Better automatic evaluation metrics for these categories should be investigated (e.g., like in VBench).
3. More qualitative examples/analysis should be included for generalization to longer shots (e.g., 15 shots) to support the claim of "AnimeShooterGen generalizes robustly to longer sequences during testing."

**Questions:**

N/A

---

> ### Author Response · Authors · 2025-11-22
> **Response to reviewer pjXJ (W1)**
>
> **W1. The compared baselines are relatively weaker baselines (e.g., IP‑Adapter+I2V and CogVideo‑LoRA). Stronger baseline models with MLLM might also be considered.**
>
> **Response:** We appreciate the reviewer's insightful feedback. We want to first emphasis that our primary objective in this work was not to propose a state-of-the-art model that surpasses all others, but rather to introduce and validate the utility of AnimeShooter dataset and to establish a baseline for the paradigm of reference-guided multi-shot animation generation. Our comparison methods are designed for the following reasons:
> 1. Simulating prevailing workflows in storytelling and animation generation: In the fields of storytelling and animation generation, the prevailing workflow typically relies on agent-based pipelines [1][2][3] rather than end-to-end architectures. These workflows generally include: (1) using customized T2I models to generate keyframes followed by I2V conversion, or (2) stitching together single shots generated by customized T2V models. To reflect this reality, we selected "IP-Adapter + I2V" and "CogVideo-LoRA" as representative baselines for these respective strategies. We prioritized open-source components over proprietary commercial models to ensure that our comparisons focused on strategies rather than the raw performance differences of underlying foundation models.
> 2. Validating the efficacy of AnimeShooter dataset: AnimeShooterGen model and CogVideo-LoRA utilize the same pretrained diffusion model and LoRA fine-tuning strategy, and the critical difference lies in the training data and the multi-shot architecture. The results demonstrated that the hierarchical annotations within our dataset provide the necessary signal for models to learn and maintain narrative and visual consistency.
>
> Moreover, to the best of our knowledge, there are currently no open-source models capable of performing consistent, reference-guided multi-shot animation generation. While we are aware of multi-shot works such as Pandora [4], Owl-1 [5], LCT [6], PMT2V [7], and MovieDreamer [8], these models either lack support for reference-guided customization, focus narrowly on human facial regions, or remain closed-source due to commercial or copyright restrictions. Since reference-guided multi-shot animation generation represents a novel paradigm, there was no off-the-shelf predecessor available for direct comparison. Consequently, we referenced the architectural designs of models like Pandora and extended them into a reference-guided framework to build AnimeShooterGen. This model is intended to serve as a baseline to validate the effectiveness of our dataset, filling the void where no accessible solution previously existed. **If there are any suitable open-source models that we may have overlooked, we would be grateful for the recommendation and are more than willing to incorporate them into our experiments.**
>
> > [1] VideoStudio: Generating Consistent-Content and Multi-scene Videos
>
> > [2] Anim-Director: A Large Multimodal Model Powered Agent for Controllable Animation Video Generation
>
> > [3] Automated Movie Generation via Multi-Agent CoT Planning
>
> > [4] Pandora: Towards General World Model with Natural Language Actions and Video States
>
> > [5] Owl-1: Omni World Model for Consistent Long Video Generation
>
> > [6] Long Context Tuning for Video Generation
>
> > [7] EchoShot: Multi-Shot Portrait Video Generation
>
> > [8] MovieDreamer: Hierarchical Generation for Coherent Long Visual Sequence

---

> ### Author Response · Authors · 2025-11-22
> **# Response to reviewer pjXJ (W2, W3)**
>
> **W2. The evaluation metrics with CLIP and DreamSim cannot capture some video quality aspects like motion smoothness. Better automatic evaluation metrics for these categories should be investigated (e.g., like in VBench).**
>
> **Response:** We thank the reviewer for this insightful comment. We have supplemented evaluation results on VBench in Appendix C.8 (Video Quality). Following table (Table 5 in manuscript) presents the evaluation results on VBench. Although IP-Adapter + I2V employs the more powerful CogVideo-5B model, it does not demonstrate superior video quality, likely due to the domain gap between real-world and animation. In contrast, CogVideo-LoRA and AnimeShooterGen which are trained on animation dataset achieve comparable or even better quality with only a 2B model, and AnimeShooterGen achieves the best performance across most metrics.
>
> | Model | Subject Cons. | Bg Cons. | Motion Smth. | Dyn. Degree | Aesthetic | Imaging |
> | ----- | ----- | ----- | ----- | ----- | ----- | ----- |
> | IP-Adapter + I2V | 0.9341 | **0.9543** | 0.9873 | 0.6500 | 0.5959 | 0.6073 |
> | Cogvideo-LoRA    | 0.9330 | 0.9539 | 0.9904 | 0.7025 | 0.6171 | **0.6524** |
> | AnimeShooterGen  | **0.9366** | 0.9517 | **0.9921** | **0.7124** | **0.6255** | 0.6512 |
>
> **W3. More qualitative examples/analysis should be included for generalization to longer shots (e.g., 15 shots) to support the claim of "AnimeShooterGen generalizes robustly to longer sequences during testing."**
>
> **Response:** We thank the reviewer for this suggestion. We have supplemented qualitative results in Appendix C.6 (Model Performance on Longer Storylines) to show the consistency robustness of the method in temporal extension and other challenges arising from long sequences.

---

### Official Review · Reviewer_ZmJj · 2025-11-01

**Soundness:** 2
**Presentation:** 4
**Contribution:** 3
**Rating:** 4
**Confidence:** 3

**Summary:**

This paper proposes AnimeShooter to address the current limitations in multi-shot datasets, which mainly focus on real-world scenarios and lack reference images. AnimeShooter is a reference-guided multi-shot animation dataset. For each shot, the dataset provides annotations of the scene and characters, as well as visual descriptions in both narrative and descriptive forms. Additionally, a subset with synchronized audio annotations, AnimeShooter-audio, is provided. Moreover, this paper introduces a reference-image-guided multi-shot video generation model based on MLLMs and diffusion models. The effectiveness of the proposed model is validated through both qualitative and quantitative experiments.

**Strengths:**

1. The paper introduces a multi-shot video dataset in the anime domain, along with a subset containing audio data, laying a foundation for advancing research in animation storytelling.
2. The paper is well-organized and highly readable, with figures concisely illustrating the workflow, data structure, and qualitative comparisons.

**Weaknesses:**

1. In the proposed method, visual information such as reference images and different shot contexts is encoded by the MLLM and aligned with the text embeddings of the diffusion model. This can lead to the loss of fine details from the reference images or shot scenes. As shown in Figure 5, without LoRA enhancement, the consistency of details is not satisfactory. However, many real-world workflows prefer models that do not require additional fine-tuning.

2. The baseline methods compared in this paper are not specifically designed for multi-shot video generation. Since they lack any cross-shot perception capability, it is unsurprising that the proposed method shows improvements over these baselines.

3. The videos generated in this paper exhibit weak storytelling and scene transitions across different shots. In my view, they resemble a combination of multiple videos rather than multiple shots of a single coherent video.

4. The paper lacks more ablation studies to demonstrate the effectiveness of the proposed model architecture.

**Questions:**

The proposed dataset contains multi-character shots, but the results presented in the paper are all single-character. Is the method capable of generating multi-character videos?

---

> ### Author Response · Authors · 2025-11-22
> **Response to reviewer ZmJj (W1)**
>
> **W1. In the proposed method, visual information such as reference images and different shot contexts is encoded by the MLLM and aligned with the text embeddings of the diffusion model. This can lead to the loss of fine details from the reference images or shot scenes. As shown in Figure 5, without LoRA enhancement, the consistency of details is not satisfactory. However, many real-world workflows prefer models that do not require additional fine-tuning.**
>
> **Response:** We appreciate this insightful observation regarding the detail preservation and the convenience of tuning-free models. We address this concern from three perspectives:
> 1. Training efficiency and computational feasibility: Our design choice was primarily driven by computational constraints and the state of the field at the time of development. There were no open-source video foundation models possessing inherent open-domain customization capabilities that we can utilize to post-train. And our extensive preliminary experiments, corroborated by existing works on single-shot customization, revealed that endowing a standard video model (like CogVideo) with general zero-shot customization capabilities is non-trivial and computationally prohibitive. Consequently, inspired by prior works such as DreamRunner[1] and MovieAgent[2], we adopted test-time fine-tuning via LoRA as a pragmatic and efficient alternative to ensure high fidelity.
>
> 2. The role of MLLM in semantic consistency: While we acknowledge that MLLM encodings may abstract away fine-grained pixel details, our experiments demonstrate that they capture indispensable high-level semantic consistency. In our comparison with CogVideo-LoRA (which employs the same pre-trained diffusion model and LoRA fine-tuning but lacks the MLLM condition), AnimeShooterGen exhibits significantly superior performance in maintaining character identity and cross-shot stylistic coherence. This indicates that the MLLM provides essential high-level semantic guidance, which effectively directs the LoRA module to focus on fine-grained alignment.
>
> 3. Dataset contribution for future tuning-free models: We emphasize that the primary contribution of this paper is the AnimeShooter dataset, with AnimeShooterGen serving as a baseline to validate its utility. Prior to this work, large-scale public datasets supporting reference-guided customization were virtually non-existent, forcing researchers to rely on internal datasets. By releasing AnimeShooter, we aim to provide the community with the necessary data foundation to train more robust, open-domain customization models (e.g., a "Video IP-Adapter") that can eventually eliminate the need for test-time fine-tuning.
>
> > [1] DREAMRUNNER: Fine-Grained Storytelling Video Generation with Retrieval-Augmented Motion Adaptation
>
> > [2] Automated Movie Generation via Multi-Agent CoT Planning

---

> ### Author Response · Authors · 2025-11-22
> **Response to reviewer ZmJj (W2)**
>
> **W2. The baseline methods compared in this paper are not specifically designed for multi-shot video generation. Since they lack any cross-shot perception capability, it is unsurprising that the proposed method shows improvements over these baselines.**
>
> **Response:** We appreciate the reviewer's insightful feedback. We want to first emphasis that our primary objective in this work was not to propose a state-of-the-art model that surpasses all others, but rather to introduce and validate the utility of AnimeShooter dataset and to establish a baseline for the paradigm of reference-guided multi-shot animation generation. Our comparison methods are designed for the following reasons:
> 1. Simulating prevailing workflows in storytelling and animation generation: In the fields of storytelling and animation generation, the prevailing workflow typically relies on agent-based pipelines [1][2][3] rather than end-to-end architectures. These workflows generally include: (1) using customized T2I models to generate keyframes followed by I2V conversion, or (2) stitching together single shots generated by customized T2V models. To reflect this reality, we selected "IP-Adapter + I2V" and "CogVideo-LoRA" as representative baselines for these respective strategies. We prioritized open-source components over proprietary commercial models to ensure that our comparisons focused on strategies rather than the raw performance differences of underlying foundation models.
> 2. Validating the efficacy of AnimeShooter dataset: AnimeShooterGen model and CogVideo-LoRA utilize the same pretrained diffusion model and LoRA fine-tuning strategy, and the critical difference lies in the training data and the multi-shot architecture. The results demonstrated that the hierarchical annotations within our dataset provide the necessary signal for models to learn and maintain narrative and visual consistency.
>
> Moreover, to the best of our knowledge, there are currently no open-source models capable of performing consistent, reference-guided multi-shot animation generation. While we are aware of multi-shot works such as Pandora [4], Owl-1 [5], LCT [6], PMT2V [7], and MovieDreamer [8], these models either lack support for reference-guided customization, focus narrowly on human facial regions, or remain closed-source due to commercial or copyright restrictions. Since reference-guided multi-shot animation generation represents a novel paradigm, there was no off-the-shelf predecessor available for direct comparison. Consequently, we referenced the architectural designs of models like Pandora and extended them into a reference-guided framework to build AnimeShooterGen. This model is intended to serve as a baseline to validate the effectiveness of our dataset, filling the void where no accessible solution previously existed. **If there are any suitable open-source models that we may have overlooked, we would be grateful for the recommendation and are more than willing to incorporate them into our experiments.**
>
> > [1] VideoStudio: Generating Consistent-Content and Multi-scene Videos
>
> > [2] Anim-Director: A Large Multimodal Model Powered Agent for Controllable Animation Video Generation
>
> > [3] Automated Movie Generation via Multi-Agent CoT Planning
>
> > [4] Pandora: Towards General World Model with Natural Language Actions and Video States
>
> > [5] Owl-1: Omni World Model for Consistent Long Video Generation
>
> > [6] Long Context Tuning for Video Generation
>
> > [7] EchoShot: Multi-Shot Portrait Video Generation
>
> > [8] MovieDreamer: Hierarchical Generation for Coherent Long Visual Sequence

---

> ### Author Response · Authors · 2025-11-22
> **Response to reviewer ZmJj (W3, W4, Q1)**
>
> **W3. The videos generated in this paper exhibit weak storytelling and scene transitions across different shots. In my view, they resemble a combination of multiple videos rather than multiple shots of a single coherent video.**
>
> **Response:** In cinematic storytelling, a "cut" implies a distinct change in angle, lighting, or background, which naturally makes shots look visually distinct compared to a single continuous "long take". Firstly, to better illustrate that our AnimeShooter dataset was designed for multi-shot, we have supplemented Appendix A.4 (Dataset Example) with a concrete example. This example features a narrative that unfolds across different scenes. Importantly, even within a single, continuous scene such as the "Birthday Table," the shots are defined by cuts that shift the camera's focus between the cake, the character Luna, and her father.
>
> Secondly, regarding the generated results, we respectfully posit that they are more than a simple combination of unrelated videos. For instance, in the right-hand example in Figure 4, the first three shots clearly depict the same young girl in the same environment—a snowy lakeside forest. Key background elements, like the morphology of the trees and the presence of snow, are preserved across these shots, demonstrating a clear example of multi-shot continuity.
>
> We concede that achieving the stricter form of "multi-shot" continuity seen in Appendix A.4 example, where background objects remain perfectly static and consistent as the camera cuts between subjects within the exact same space, remains a formidable challenge for current generative models. This level of object permanence and spatial awareness often necessitates more advanced techniques, such as explicit memory architectures or 3D methods. Therefore, in this paper, our goal is narrative coherence (same character, sequential logic) across these cuts, rather than the visual continuity of a morphing scene.
>
> **W4. The paper lacks more ablation studies to demonstrate the effectiveness of the proposed model architecture.**
>
> **Response:** Thank you for your suggestion. We have supplemented Appendix C.9 (Ablation Studies) to show the ablation studies. Following table (Table 6 in manuscript) presents the results of ablation study, which includes the following configurations: (1) w/o MLLM condition, where the MLLM condition is replaced with the original text condition (i.e., CogVideo-LoRA); (2) w/o LoRA Enhancement, where no LoRA fine-tuning based on a specific IP, and zero-shot capability is tested directly; and (3) w/o Reference Image, where the reference image in the MLLM is replaced with a blank image. The results further confirm that after training on the AnimeShooter dataset, the MLLM is able to capture high-level semantic consistency. Replacing it with the diffusion model’s original text condition or using a blank image significantly degrades model performance. Moreover, the results highlight the importance of the detail alignment achieved during the LoRA Enhancement stage, and the high-level semantic consistency provided by the MLLM cannot be fully realized at the pixel level without it.
>
> | Automatic Metrics (CLIP$\uparrow$) | Shot1 | Shot2 | Shot3 | Shot4 | Mean | HarMeanP |
> | ----- | ----- | ----- | ----- | ----- | ----- | ----- |
> | w/o MLLM condition   | 0.7297 | 0.7200 | 0.7417 | 0.7413 | 0.7332 |0.5028|
> | w/o LoRA Enhancement | 0.7628 | 0.7430 | 0.7376 | 0.7362 | 0.7449 |0.5280|
> | w/o Reference Image  | 0.7065 | 0.7167 | 0.7208 | 0.7264 | 0.7176 |0.4934|
> | AnimeShooterGen | **0.8022** | **0.7949** | **0.7970** | **0.7986** | **0.7982** | **0.6121** |
>
> | Automatic Metrics (DreamSim$\downarrow$) | Shot1 | Shot2 | Shot3 | Shot4 | Mean | HarMeanP |
> | ----- | ----- | ----- | ----- | ----- | ----- | ----- |
> | w/o MLLM condition   | 0.4777 | 0.5060 | 0.4842 | 0.4864 | 0.4886 |0.7759|
> | w/o LoRA Enhancement | 0.4773 | 0.5198 | 0.5178 | 0.5323 | 0.5118 |0.7940|
> | w/o Reference Image  | 0.5363 | 0.5348 | 0.5213 | 0.5131 | 0.5264 |0.7926|
> | AnimeShooterGen | **0.3484** | **0.3820** | **0.3799** | **0.3764** | **0.3717** | **0.6413** |
>
> **Q1. The proposed dataset contains multi-character shots, but the results presented in the paper are all single-character. Is the method capable of generating multi-character videos?**
>
> **Response:** Thanks for your question. Our current framework is designed for single-IP and does not support multi-IP generation. The model accepts only a single reference image per generation instance. Severe problems including attribute conflation[1] may occur if directly applying single-IP models to multi-IP generation.
>
> Recognizing its importance, AnimeShooter provides multi-IP annotations and reference image masks for each video, hoping to advance future multi-IP customization research.
>
> > [1] ConceptMaster: Multi-Concept Video Customization on Diffusion Transformer Models Without Test-Time Tuning

---

### Official Review · Reviewer_pA6S · 2025-11-02

**Soundness:** 1
**Presentation:** 3
**Contribution:** 3
**Rating:** 6
**Confidence:** 3

**Summary:**

This paper introduces AnimeShooter, a large-scale, reference-guided multi-shot animation dataset featuring hierarchical story- and shot-level annotations, synchronized audio tracks, and strong cross-shot visual consistency. To demonstrate its utility, the authors propose AnimeShooterGen, a baseline framework that integrates a Multimodal Large Language Model (MLLM) with a video diffusion model for generating coherent multi-shot animations.

**Strengths:**

AnimeShooter provides structured, story-aware, and reference-guided annotations, filling a significant gap in current video-generation datasets. Clear separation between story-level and shot-level elements enables both global narrative control and local visual coherence.
The open release of both dataset and baseline has high potential to become a standard benchmark for multi-shot animation generation.

**Weaknesses:**

1.Hierarchical captioning reduces drift but lacks visual grounding, leaving potential hallucination issues.
2.Only basic normalization is used; no explicit domain alignment across different animation styles and for object segmentation methods, fine-tuning on ~500 frames offers limited adaptation from real-world to animated content.
3.For keyframe-selection, real-video heuristics are applied; not well-suited for low frame-rate animation.
4.No explicit loss or alignment; character consistency relies on semantic coincidence. Lacks global temporal structure, leading to potential drift over long multi-shot sequences.
5.The MLLM–diffusion pipeline is computationally heavy, with no reported efficiency metrics or optimization strategy.

**Questions:**

see weaknesses

---

> ### Author Response · Authors · 2025-11-22
> **Response to reviewer pA6S (W1, W2, W3)**
>
> **W1. Hierarchical captioning reduces drift but lacks visual grounding, leaving potential hallucination issues.**
>
> **Response:** We thank the reviewer for this insightful comment. We agree that ensuring strong visual grounding and mitigating potential hallucination are critical for dataset quality, so we have implemented a multi-faceted strategy to address this challenge:
> 1. Leveraging SOTA MLLMs: We employed Gemini 2.0 Flash for AnimeShooter (as version 2.5 was not yet available at that time) and Gemini 2.5 Pro for AnimeShooter-audio. The Gemini family represents the state-of-the-art in dense video understanding and annotation, and their advanced capabilities serve as the foundational defense against hallucination. This choice is consistent with best practices in the field, as seen in related works like LCT [1], which also utilize Gemini 1.5 for generating both global and shot-level video descriptions. Furthermore, our empirical analysis indicated that the model's annotation fidelity is highest for videos under one minute. We therefore pre-processed long videos into ~1-minute clips to ensure the MLLM operates within its optimal performance window, maximizing the quality of visual grounding.
> 2. Optimized hierarchical top-down strategy: Our choice of a top-down hierarchical strategy was based on a comparative analysis against a bottom-up approach. The bottom-up method annotates short clips before aggregating them. This method avoids effect from hierarchical captioning since each shot is annotated independently, but was found to be prone to significant character identity drift and narrative discontinuities. In contrast, our top-down method first establishes a global narrative context, which then acts as a powerful constraint on the generation of subsequent shot-level descriptions to enforce narrative consistency. The prompts used in this process were also iteratively refined to meet stringent quality standards.
> 3. Human verification checkpoints: Crucially, as stated on lines 231-232, we integrated human verification checkpoints into our pipeline. We validated the accuracy of the generated story scripts and reference images on a representative subset of the data. This process confirmed that instances of significant hallucination are rare and that the annotations maintain high fidelity to the source videos.
>
> > [1] Long Context Tuning for Video Generation
>
>
> **W2. Only basic normalization is used; no explicit domain alignment across different animation styles and for object segmentation methods, fine-tuning on ~500 frames offers limited adaptation from real-world to animated content.**
>
> **Response:** To ensure our response is as precise as possible, we interpret the concern about "domain alignment" and "adaptation" as primarily relating to the potential domain gap when using the real-world-trained Sa2VA model for character segmentation in animation. If our understanding is correct, our approach is as follows. If we have missed a nuance, we would be grateful for your further clarification.
>
> While we did not use explicit domain alignment, during our extensive data processing, we observed that the Sa2VA model demonstrated remarkably robust segmentation capabilities on animated content, effectively handling the majority of scenes in our dataset. To further ensure the quality of the reference images derived from Sa2VA, we implemented a multi-stage refinement and validation process:
> 1. Model-based validation: The refined masks after morphological operations are then validated by the powerful InternVL model. This step serves as a stringent quality gate, automatically discarding any substandard segmentations that fail to meet our predefined structural and semantic coherence criteria.
> 3. Manual quality monitoring: We also conducted manual inspections on a representative subset (including diverse animation styles) of the data. This human oversight allowed us to continuously monitor and optimize the entire pipeline, ensuring high-quality segmentations.
>
> This combination of a segmentation model with strong inherent generalization capabilities, coupled with our validation and refinement protocol, effectively mitigated the domain shift concerns for the purpose of generating high-quality reference data.
>
>
> **W3. For keyframe-selection, real-video heuristics are applied; not well-suited for low frame-rate animation.**
>
> **Response:** We wish to clarify that the frame sampling step is a specific component within our reference image generation pipeline (Section 3.3), not a generic keyframe extraction process for real-world video. For each character, we first leverage the pre-extracted story scripts to identify all shots in which they appear. From this targeted set of shots, we sample candidate frames at a consistent rate of 1 fps. This rate was not an arbitrary choice derived from real-video heuristics; rather, it was a hyperparameter deliberately selected and optimized based on empirical analysis of our animation dataset.

---

> ### Author Response · Authors · 2025-11-22
> **Response to reviewer pA6S (W4, W5)**
>
> **W4. No explicit loss or alignment; character consistency relies on semantic coincidence. Lacks global temporal structure, leading to potential drift over long multi-shot sequences.**
>
> **Response:** We respectfully argue that the character consistency in our framework is derived from robust, high-level semantic representations explicitly learned by the MLLM, rather than "semantic coincidence." During the Single-Shot Training phase, the MLLM is optimized to extract high-level visual attributes from the reference image and integrate them into the diffusion condition, a capability that is further consolidated during the Multi-Shot Training phase. This is evidenced by our ablation study in Section 5.4 (Figure 5), where the model maintains consistency in character appearance and artistic style even without the final LoRA enhancement stage, demonstrating that the MLLM provides a fundamental semantic anchor independent of pixel-level overfitting. Furthermore, our comparison with CogVideo-LoRA—which utilizes the same diffusion backbone and LoRA fine-tuning but lacks the MLLM condition—reveals that the MLLM condition is the decisive factor for identity preservation.
>
> Regarding the concern about temporal structure, our experiments demonstrate that the proposed autoregressive mechanism generalizes effectively to long sequences without significant drift. While the model is trained on sequences of only 3 consecutive shots, our primary evaluations are conducted on 4-shot sequences, showcasing its immediate ability to maintain coherence beyond the training horizon. We further validate long-term robustness in Appendix C.6 (Model Performance on Longer Storylines) by extending generation to 15-shot storylines. The quantitative and qualitative results show that the model sustains high performance in character consistency and contextual consistency across these longer sequences.
>
>
> **W5. The MLLM–diffusion pipeline is computationally heavy, with no reported efficiency metrics or optimization strategy.**
>
> **Response:** We thank the reviewer for this suggestion. We have reported training costs and inference efficiency in Appendix B.2 (Implementation Details): (1) All training stages are implemented on GPUs with 40G memory, and the computational costs for the different training stages are: 1400 GPU hours for Condition Alignment, 900 GPU hours for Single-Shot Training, 400 GPU hours for Multi-Shot Training, and 20 GPU hours for the final LoRA Enhancement per IP. (2) Inference for a 4-shot story takes approximately 720 seconds and can be run on a single GPU with 40GB of memory.
>
> Regarding optimization, we leveraged memory-saving techniques such as DeepSpeed for training (enabling 8×40GB GPU support) and pipeline offloading for inference (enabling 1×40GB GPU deployment) to minimize memory overhead. The current inference time is predominantly limited by the base CogVideo model, which necessitates 50 denoising steps. Since AnimeShooterGen serves as the baseline for the reference-guided multi-shot paradigm, this study prioritizes generation consistency over aggressive speed acceleration. We consider efficiency optimization (e.g., via model distillation) to be a separate track of research that is better suited for future work once this paradigm is fully established.

---

### Author Response · Authors · 2025-12-03
**Summary of Reviews and Rebuttals**

We sincerely thank the ACs and all reviewers for their time, effort, and constructive feedback. Below we summarize the reviews and how our rebuttal addresses the main concerns.

## Strengths:
We are greatly encouraged that the reviewers recognized the value of our work and identified our contributions to the field of storytelling and animation generation:

**Addressing a critical research gap.** AnimeShooter establishes a novel paradigm of reference-guided multi-shot animation generation, effectively filling a critical void in both datasets and models within the current landscape. The reviewers commended AnimeShooter for filling a "significant gap" in current datasets and isolating a "genuinely under‑served setting" compared to existing datasets. [Reviewers pA6S, pjXJ, Kftk]

**The dataset is comprehensive and open-sourced.** The reviewers commended the value of the AnimeShooter dataset for the community, particularly its comprehensive hierarchical annotations for storyboard, specific character reference images essential for customization tasks, and the inclusion of a subset with synchronized audio annotations. [Reviewers pA6S, ZmJj, pjXJ, Kftk]

**AnimeShooterGen setting an effective baseline.** The reviewers acknowledged the proposed AnimeShooterGen as a technically sound framework and a strong baseline model. It effectively demonstrates the dataset's validity and establishes a standard benchmark for evaluating challenging metrics such as cross-shot visual consistency and reference adherence. [Reviewers pA6S, pjXJ, Kftk]

## Concerns:

### Dataset:

#### 1. Concern about dataset quality
Reviewer pA6S raised concerns regarding potential hallucination issues in MLLM-generated annotations and the domain bias for real-world scenarios. We addressed these concerns by detailing our iteratively refined pipeline and rigorous quality control protocol, especially the human verification checkpoints on a representative subset covering diverse animation styles.

### Model & Experiments:

We want to first emphasize that AnimeShooterGen is designed primarily as a baseline model to validate the utility of our dataset and fill the void in the reference-guided multi-shot animation landscape. As we acknowledge significant potential for future improvements upon this baseline, there are no direct open-source predecessors.

#### 1. Rationale for LoRA enhancement

Reviewers ZmJj and Kftk questioned the zero-shot performance without LoRA enhancement. We added experiments to show AnimeShooterGen's zero-shot capabilities and clarified that it cannot currently serve as a general "Video IP-Adapter", as the MLLM provides high-level semantic consistency without details. Due to the lack of open-source video foundation models with inherent customization capabilities and the prohibitive computational cost of training general zero-shot capabilities from scratch, we adopted LoRA as a pragmatic strategy. Crucially, we also highlighted that the release of the AnimeShooter dataset provides the necessary data foundation for the community to train robust, open-domain customization models in the future.

#### 2. Justification of Comparative Models

Reviewer ZmJj and pjXJ noted that the comparative models were not specifically designed for multi-shot generation. To the best of our knowledge, there are currently no open-source models capable of performing consistent, reference-guided multi-shot animation generation, so we designed comparison methods for the following reasons: (1) simulating prevailing workflows; (2) as the most important ablation study, validating that the superior consistency and narrative coherence are derived from our dataset.

#### 3. Expanded Ablation Studies and Analysis

Incorporating reviewers' suggestions, we have supplemented the manuscript with expanded experiments including:
- Ablation studies in Appendix C.9. [Reviewers ZmJj: W4; Kftk: W1, W2]
- VBench metrics in Appendix C.8. [Reviewers pjXJ: W2]
- Qualitative examples of longer shots in Appendix C.6. [Reviewers pjXJ: W3]
- Visual context frames in Appendix C.10. [Reviewer Kftk: W3]

### Motivation:

#### 1. Definition of Multi-shots

The reviewer ZmJj is concerned that our work resembles a combination of multiple videos rather than multiple shots of a single coherent video. To better clarify that AnimeShooter is designed for multi-shot settings, we have updated Appendix A.4 with a concrete sequence unfolding across different scenes. Within one continuous scene, shots are defined by cuts that shift the camera’s focus between the cake, Luna, and her father. Regarding the generated results, we respectfully argue that they go beyond a simple concatenation of unrelated videos. For example, in the right-hand case of Figure 4, the first three shots depict the same young girl in the same snowy lakeside forest. Key background elements are consistently preserved, illustrating clear multi-shot continuity.

---

### Meta-Review · Area_Chair_4Ccn · 2026-01-14

**Summary:**

This paper introduces AnimeShooter, a large-scale, reference-guided multi-shot animation dataset designed to address the lack of structured, animation-focused datasets for coherent multi-shot video generation. Each animation “story” contains hierarchical annotations at both the story level (e.g., storyline, main characters with reference images, and scenes) and the shot level (e.g., ordered shots with scene, character, and narrative/descriptive captions), with an additional subset providing synchronized audio annotations. The dataset is constructed via an automated pipeline that collects animation videos and generates high-quality annotations and reference images using large multimodal models. To demonstrate its utility, the paper proposes AnimeShooterGen, a baseline framework that integrates an MLLM with a video diffusion model to generate visually consistent multi-shot animations conditioned on reference images, textual descriptions, and previously generated shots.

**Reviewer Concerns:**

Across the reviews, the main concerns center on limited visual grounding and character consistency, heavy reliance on test-time LoRA fine-tuning, and insufficiently strong experimental validation. Reviewers question whether the method truly generalizes beyond few-shot IP customization, noting potential hallucination, loss of fine-grained visual details, weak global temporal structure, and limited storytelling coherence across shots. In addition, the evaluation is considered incomplete due to weak or mismatched baselines, insufficient ablation studies (e.g., with/without LoRA, zero-shot settings), lack of efficiency analysis, and metrics that do not fully capture video quality aspects such as motion smoothness and long-range coherence.

**Reviewer Scores:**

Overall, the reviewers’ ratings are generally around the borderline between marginal acceptance and marginal rejection. During the rebuttal period, although the authors made efforts to address several major concerns, such as clarifying the model’s performance in a zero-shot setting, some reviewers explicitly stated that their original scores already reflected their assessment of the paper and therefore chose to maintain their ratings without change.

---

### Decision · Program_Chairs · 2026-01-26

Reject